# Convex Dominance in Deep Learning I: A Scaling Law of Loss and Learning Rate

**Zhiqi Bu**

FAIR, Meta Superintelligence Labs

zhiqibu@meta.com

**Shiyun Xu**

Independent Researcher

shiyunxulara@gmail.com

**Jialin Mao**

Independent Researcher

jialinmao000@gmail.com

## Abstract

Deep learning has non-convex loss landscape and its optimization dynamics is hard to analyze or control. Nevertheless, the dynamics can be empirically convex-like across various tasks, models, optimizers, hyperparameters, etc. In this work, we examine the applicability of convexity and Lipschitz continuity in deep learning, in order to precisely control the loss dynamics via the learning rate schedules. We illustrate that deep learning quickly becomes weakly convex after a short period of training, and the loss is predicable by an upper bound on the last iterate, which further informs the scaling of optimal learning rate. Through the lens of convexity, we build scaling laws of learning rates and losses that extrapolate as much as $80\times$ across training horizons and $70\times$ across model sizes.

## 1 Introduction

Deep learning has highly non-convex and complex loss landscape, e.g. the global minimum may not be unique, and there may be many local minima and saddle points where the optimization can be trapped (Garipov et al., 2018; Choromanska et al., 2015; Dauphin et al., 2014; Jin et al., 2017; Sagun et al., 2016). Nevertheless, the empirical success of optimization in deep learning has implied that some benign properties may hold and be leveraged.

In fact, there has been a long history of empirically exploring convexity in deep learning. For example, Llama training (non-convex) with AdamW (Loshchilov & Hutter, 2019) is closely similar to convex optimization with SGD, in terms of the shapes of loss curves in Figure 1 of (Schaipp et al., 2025). Empirical evidence by (Zhou et al., 2019) has shown that SGD follows a star-convex path during the optimization of neural networks. In addition, convexity is universally observed along the direction of gradients in vision and language models by (Bu & Xu, 2025).

Another line of research has focused on the two-dimensional loss landscape, by (Li et al., 2018; Im et al., 2016) (along two random and normalized directions) and (Allen-Zhu et al., 2019b) (along the gradient and the negatively curved direction of the Hessian), which is approximately locally convex for residual neural networks with various depth and width. Furthermore, these convex-like loss landscapes are also observed on large language models such as RoBERTa, LLaMA, Qwen, and Mistral in (Zhong et al., 2022; Chen et al., 2025; Lee et al., 2024).

Besides these low-dimensional loss landscapes, the Hessian spectrum provides a rigorous local notion of curvature: a positive-definite Hessian (all eigenvalues positive) indicates that the loss landscape is locally convex around that point. Empirical observations show that at initialization, the Hessian often contains many large negative eigenvalues, which quickly shift toward zero and become much smaller in magnitude than the positive eigenvalues. Consequently, the spectrum becomes dominated by positive eigenvalues and the loss landscape becomes approximately convex (Yao et al., 2020; Papyan, 2018; Sankar et al., 2021; Zhang et al., 2024).

In addition, theoretical analysis shows that two-layer and deeper neural networks can be regarded as convex in the wide-network limit, under neural tangent kernel (Jacot et al., 2018; Lee et al., 2019; Du et al., 2018; 2019; Li & Liang, 2018; Allen-Zhu et al., 2019a), neural repopulation (Fang et al., 2019; 2022), and mean field (Mei et al., 2018; Chizat & Bach, 2018).

As a matter of fact, many works in deep learning have been drawing useful understanding from convex analysis to non-convex deep learning: Sutskever et al. (2013) shows that momentum significantly accelerates the convergence; Defazio et al. (2023) explains the effectiveness of learning rate warmup and decay from a convex viewpoint; the effect of weight decay (Krogh & Hertz, 1991) is first understood in ridge regression (Hoerl & Kennard, 1970).

Specifically,the relationship between the learning rate sequence $\{\eta_t\}$ and the upper bounds of loss sequence $\{L_t\}$ has been heavily studied. These bounds come from different settings, studying convex or strongly convex loss, Lipschitz continuous or smooth loss, finite-iteration or asymptotic bound, averaged or last iterate, etc. Our work is directly based on Corollary 12 of (Defazio et al., 2023) (re-stated in (2.4)), and a simplified version can be found in (2.3), which already provides some insights as we will summarize in Example 2.3.

## 1.1 RELATED WORK

**Convex to deep learning.** It is well-known in convex regime that loss can converge at $O(1/\sqrt{T})$ and optimal learning rate is $O(1/\sqrt{T})$. However, whether or when these conclusions hold in deep learning is largely unclear. From this perspective, our work is most closely related to (Schaipp et al., 2025), both following from (Defazio et al., 2023). In contrast, we not only extend major findings of (Schaipp et al., 2025) theoretically (e.g. our qualifying exam in Condition 2.5 covers any schedule), but also focus more on empirical validation. For example, $O(1/\sqrt{T})$ convergence of loss and consistent patterns across models and optimizers in our Figures 6-12 are not presented in (Schaipp et al., 2025), which focuses on one model and single training horizon like our Figures 2-5. In particular, our empirical approach is data-driven and practically applicable in large-scale deep learning.

**Scaling laws.** Current popular scaling laws are primarily about loss (Kaplan et al., 2020; Hoffmann et al., 2022) (i.e. scaling laws of loss), predicting how loss changes as model sizes and training horizons change, assuming optimal learning rate. As a result, learning rate is not explicitly presented in these laws. Other laws (i.e. scaling laws of learning rate) can scale learning rate across training horizons $\eta_{\text{peak}}^* = \lambda T^{-\alpha}$, where $\alpha = 0.125$ in (Bi et al., 2024), $\alpha \in \{0.32, 0.38, 0.42, 0.65, 0.70\}$ by Table 5 in (Bjorck et al., 2024), and some are horizon-unaware in (Porian et al., 2024; Wang et al., 2024) (i.e. $\alpha = 0$). Nevertheless, these laws may not predict loss. In this work, we propose a law that simultaneously predict loss and optimal learning rate for the fixed value $\alpha = 0.5$.

**Learning rate transfer.** Maximal update parameterization (muP) (Yang et al., 2022) is a technique to transfer optimal learning rate across model size. While Table 1 of (Yang et al., 2022) claimed it also transfers across training horizons, this contradicts with empirical evidence in (Bjorck et al., 2024) (see Section 3.3) and our analysis.

## 1.2 CONTRIBUTIONS

In this work, we study the scaling law of deep learning loss and learning rate, through the lens of convex loss (non-smooth) and bounded gradient. We will establish a series of generalizations from convex theory to deep learning, which is presented in Table 3 and summarized as follows.

1. We study the convex-like behaviors in deep learning for general model architectures, optimizers and learning rate schedules, hence establishing a non-asymptotic mapping from learning rate sequence to loss sequence.

2. We generalize to an asymptotic upper bound of loss, achieving $O(1/\sqrt{T})$ convergence when (I) the peak learning rate is scaled by $1/\sqrt{T}$ and (II) the learning rate schedule is qualified.

3. We propose a data-driven method to fit the asymptotic bound, establishing a scaling law across training horizons and model sizes.

## 2 CONVERGENCE OF SGD UNDER CONVEX LOSS

### 2.1 REVISITING NON-ASYMPTOTIC BOUND OF SGD

We consider the stochastic gradient descent (SGD) as $\mathbf{w}_{t+1} = \mathbf{w}_t - \eta_{t+1}\mathbf{g}(\mathbf{w}_t)$, where $\mathbf{w}$ is the parameters, $\eta$ is the learning rate, $\mathbf{g}$ is the mini-batch gradient with $\mathbb{E}\mathbf{g}(\mathbf{w}) = \nabla L$, $0 \le t < T$ is the iteration and $T$ is the training horizon (i.e. number of iterations). The learning rate is defined by two factors via $\eta_{\text{peak}} \cdot s_t(T)$: (I) the learning rate schedule, which is a function $s_t \in [0, 1]$ (e.g. linear decay is $s_t(T) = 1 - t/T$), and (II) the peak learning rate $\eta_{\text{peak}} \in \mathbb{R}^+$ which is a positive scalar.

We briefly review the convergence analysis under the convex and bounded gradient conditions.

**Condition 2.1.** *Denoting a differentiable function as $L$ and its gradient as $\nabla L$, then $L$ is*

$$\text{convex if } \forall(\mathbf{w}, \boldsymbol{x}), L(\mathbf{w}) - L(\boldsymbol{x}) \le (\mathbf{w} - \boldsymbol{x})^\top \nabla L(\mathbf{w}) \tag{2.1}$$

$$\text{bounded in gradient if } \exists G \text{ s.t. } \forall \mathbf{w}, \mathbb{E}\|\mathbf{g}(\mathbf{w})\|^2 \le G^2 \tag{2.2}$$

**Remark 2.2.** *In Condition 2.1, the convexity is not necessary for our analysis as it can be replaced by star-convexity and iterate-wise convexity along the optimization path, i.e.*

$$L(\mathbf{w}_t) - L(\mathbf{w}_*) \le (\mathbf{w}_t - \mathbf{w}_*)^\top \nabla L(\mathbf{w}_t)$$

*where $\mathbf{w}_* \in argmin_\mathbf{w} L(\mathbf{w})$ is the minimizer, and*

$$L(\mathbf{w}_t) - L(\mathbf{w}_s) \le (\mathbf{w}_t - \mathbf{w}_s)^\top \nabla L(\mathbf{w}_t).$$

*Additionally, the bounded gradient condition reduces to Lipschitz continuity when SGD is full-batch.*

In the parameter space, with $\mathbf{g}_t := \mathbf{g}(\mathbf{w}_t)$,

$$\|\mathbf{w}_{t+1} - \mathbf{w}_*\|^2 = \|\mathbf{w}_t - \mathbf{w}_*\|^2 - 2\eta_{t+1}(\mathbf{w}_t - \mathbf{w}_*)^\top \mathbf{g}_t + \eta_{t+1}^2\|\mathbf{g}_t\|^2$$

For bounded gradient and convex loss, denoting $L_* = \min_\mathbf{w} L(\mathbf{w})$, we have in expectation

$$\mathbb{E}\|\mathbf{w}_{t+1} - \mathbf{w}_*\|^2 \le \mathbb{E}\|\mathbf{w}_t - \mathbf{w}_*\|^2 - 2\eta_{t+1}(L_t - L_*) + \eta_{t+1}^2 G^2$$

Telescoping sum gives

$$0 \le \mathbb{E}\|\mathbf{w}_\tau - \mathbf{w}_*\|^2 \le \|\mathbf{w}_0 - \mathbf{w}_*\|^2 - 2\sum_{t=0}^{\tau-1}\eta_{t+1}(L_t - L_*) + \sum_{t=0}^{\tau-1}\eta_{t+1}^2 G^2$$

Dividing by $2\sum_t \eta_{t+1}$ and applying Jensen's inequality, we obtain an upper bound of $L(\bar{\mathbf{w}}_\tau)$, where $\bar{\mathbf{w}}_\tau = \frac{\sum_{t=0}^{\tau-1}\eta_{t+1}\mathbf{w}_t}{\sum_{t=0}^{\tau-1}\eta_{t+1}}$ is the averaged iterate and $D := \|\mathbf{w}_0 - \mathbf{w}_*\|$,

$$\mathbb{E}L(\bar{\mathbf{w}}_\tau) \le \frac{\sum_{t=0}^{\tau-1}\eta_{t+1}L_t}{\sum_{t=0}^{\tau-1}\eta_{t+1}} \le L_* + \frac{D^2}{2\sum_{t=1}^\tau \eta_t} + \frac{G^2\sum_{t=1}^\tau \eta_t^2}{2\sum_{t=1}^\tau \eta_t} := L_{\text{SGD-ave}}(\{\eta_t\}). \tag{2.3}$$

**Finding 2.3.** *For constant learning rate $\eta$, the loss upper bound derived from convex analysis in (2.3) simplifies to $L_* + \frac{D^2}{2T\eta} + \frac{\eta G^2}{2}$. This aligns with the empirical trade-off in deep learning that larger $\eta$ converges faster but to a higher loss, and vice versa [See Fig 10.14 (WikiDocs, 2025)]. Furthermore, this loss bound is minimized by $\eta_* = \frac{D}{\sqrt{T}G}$ in convex analysis, which underlies the fast convergence observed in deep learning such as D-adaptation (Defazio & Mishchenko, 2023), Prodigy (Mishchenko & Defazio, 2023), DoG (Ivgi et al., 2023), and DoWG (Khaled et al., 2023).*

With one extra term in (Defazio et al., 2023, Corollary 12), we have a bound of any single iterate:

$$\mathbb{E}L(\mathbf{w}_\tau) \le L_* + \frac{D^2}{2\sum_{t=1}^\tau \eta_t} + \frac{G^2\sum_{t=1}^\tau \eta_t^2}{2\sum_{t=1}^\tau \eta_t} + \frac{G^2}{2}\sum_{k=1}^{\tau-1}\frac{\eta_k}{\sum_{t=k+1}^\tau \eta_t}\frac{\sum_{t=k}^\tau \eta_t^2}{\sum_{t=k}^\tau \eta_t} \tag{2.4}$$

Note that (2.3) and (2.4) translate an arbitrary learning rate sequence $\{\eta_t\}$ to an upper bound of the loss value. While both bounds shed some insights on the loss dynamics, we focus on the bound from (2.4), since (2.3) can be less precise in characterizing the loss curves (c.f. Figure 2 and Figure 13 in (Schaipp et al., 2025)).

## 2.2 ASYMPTOTIC LOSS BOUND AT LAST ITERATION

For any training horizon $T$, we can characterize the loss at the last iteration $\mathbf{w}_T$ via (2.4). We show the upper bounds on the loss in terms of different learning rate schedules, the optimal peak learning rate $\eta_{\text{peak}}^*$, and the optimal loss bound in Table 1, where the results are derived in Theorem 1 in Appendix A.

**Table 1** Convergence of optimal loss and optimal learning rate by Theorem 1 under different schedules.

| learning rate schedule | $\eta_t$ formula | upper bound of $\mathbb{E}L(\mathbf{w}_T)$ | optimal loss bound | optimal $\eta_{\text{peak}}^*(T)$ |
|---|---|---|---|---|
| constant | $\eta_{\text{peak}}$ | $L_* + \frac{D^2}{2T\eta_{\text{peak}}} + \frac{\eta_{\text{peak}}G^2}{2}\ln T$ | $L_* + DG\sqrt{\frac{\ln T}{T}}$ | $\frac{D}{G\sqrt{\ln T \cdot T}}$ |
| square-root inverse | $\eta_{\text{peak}}/\sqrt{t}$ | $L_* + \frac{D^2}{4\sqrt{T}\eta_{\text{peak}}} + \frac{\eta_{\text{peak}}G^2\ln T}{4\sqrt{T}}$ | $L_* + DG\sqrt{\frac{\ln T}{4T}}$ | $\frac{D}{G\sqrt{\ln T}}$ |
| linear decaying | $\eta_{\text{peak}}\left(1 - t/T\right)$ | $L_* + \frac{D^2}{T\eta_{\text{peak}}} + \eta_{\text{peak}}G^2$ | $L_* + 2DG\sqrt{\frac{1}{T}}$ | $\frac{D}{G\sqrt{T}}$ |
| cosine decaying | $\eta_{\text{peak}}\frac{1+\cos(\pi t/T)}{2}$ | $L_* + \frac{D^2}{T\eta_{\text{peak}}} + \eta_{\text{peak}}G^2 \cdot 1.061$ | $L_* + 2DG\sqrt{\frac{1.061}{T}}$ | $\frac{D}{G\sqrt{1.061T}}$ |
| warmup-stable-decay | $\begin{cases}\eta_{\text{peak}} & \text{if } t < cT \\ \eta_{\text{peak}}\frac{T-t}{T-cT} & \text{if } t \geq cT\end{cases}$ | $L_* + \frac{D^2}{(1+c)T\eta_{\text{peak}}} + \eta_{\text{peak}}G^2\left[1 + \frac{1}{2}\ln\left(\frac{1+c}{1-c}\right)\right]$ | $L_* + 2DG\sqrt{\frac{1+\frac{1}{2}\ln\left(\frac{1+c}{1-c}\right)}{(1+c)T}}$ | $\frac{D}{G\sqrt{(1+c)\left(1+\frac{1}{2}\ln\left(\frac{1+c}{1-c}\right)\right)T}}$ |

We summarize some insightful observations in Table 1.

- The optimal loss convergence rate is $O(1/\sqrt{T})$ and it can be achieved by some schedules like linear, cosine decay and warmup-stable-decay (WSD or trapezoid; (Xing et al., 2018; Hägele et al., 2024)), which we term as the *qualified schedules*. In contrast, some other schedules like constant learning rate only achieve a suboptimal rate $O(\sqrt{\ln T/T})$.

- The qualified schedules have common patterns: (1) the schedules are horizon-aware, i.e. $\eta_t$ is dependent on $T$, in contrast to the constant and square-root inverse schedules; (2) the loss bounds take the following form for some constants $q_1, q_2$:

$$\mathbb{E}L(\mathbf{w}_T) \lesssim L_* + \frac{q_1^2}{T\eta_{\text{peak}}} + \eta_{\text{peak}}q_2^2 := L_{\text{SGD-last}}(\eta_{\text{peak}}, T) \tag{2.5}$$

- The optimal learning rate of a qualified schedule is $O(1/\sqrt{T})$.

While Theorem 1 have considered specific schedules and optimal peak learning rate, in what follows, we will generalize to non-optimal peak learning rates in Section 2.3 and introduce an exam to select qualified schedules in Section 2.4.

## 2.3 OPTIMAL CONVERGENCE UNDER SCALED LEARNING RATE

We show in Corollary 2.4 that any scaled $\eta_{\text{peak}}$ can achieve the optimal $O(1/\sqrt{T})$ loss convergence, regardless of whether $\eta_{\text{peak}}$ is optimal. Here, we scale $\eta_{\text{peak}}$ by dividing any reference learning rate $\eta_{\text{ref}}$ to $\sqrt{T}$.

**Corollary 2.4.** *Consider a learning rate schedule and $\eta_{\text{peak}}$ that satisfy (2.5) for SGD under Condition 2.1.*

*1. Any $\eta_{\text{ref}} \in \mathbb{R}_+$ achieves the asymptotically optimal convergence rate:*

$$L_{\text{SGD-last}}(\eta_{\text{peak}} = \eta_{\text{ref}}/\sqrt{T}, T) - L_* \sim Q(\eta_{\text{ref}})/\sqrt{T} = O\left(1/\sqrt{T}\right),$$

*in which we define $Q(\eta_{\text{ref}}) := q_1^2/\eta_{\text{ref}} + \eta_{\text{ref}}q_2^2$.*

*2. Additionally, the optimal $\eta_{\text{ref}}^* = \operatorname{argmin}_{\eta_{\text{ref}}} Q = q_1/q_2$, and $L_{\text{SGD-last}}(T) - L_* \sim \frac{2q_1 q_2}{\sqrt{T}}$.*

## 2.4 QUALIFYING EXAM FOR LEARNING RATE SCHEDULES

We *qualify* a learning rate schedule function $s_t(T)$ if (2.5) satisfies the following

$$L_{\text{SGD-last}}(\eta_{\text{peak}} = 1/\sqrt{T}, T) - L_* = O\left(1/\sqrt{T}\right),$$

where we simply choose $\eta_{\text{ref}} = 1$ as supported by Corollary 2.4 part 1.

We highlight that the qualifying exam can be conducted in a training-free way without running any model optimization. We propose to employ symbolic analysis in Condition 2.5, and the full derivation from discrete summation in (2.4) to continuous definite integrals can be found in Appendix A.2.

**Condition 2.5.** *We claim a learning rate schedule function $s_t(T) \in [0, 1]$ is qualified, if the following holds with $\eta_t(T) := s_t(T)/\sqrt{T}$:*

$$\frac{D^2}{2 \int_0^T \eta_t dt} + \frac{G^2}{2} \int_0^T \left( \frac{\eta_t^2}{\int_t^T \eta_k dk} \right) dt = O(1/\sqrt{T}).$$

Theoretically, we prove in Theorem 1 that

- linear decaying, cosine decaying, and WSD schedules pass the qualifying exam;
- constant and square-root inverse schedules fail.

Empirically, we validate the effectiveness of our qualifying exam in Figure 1.

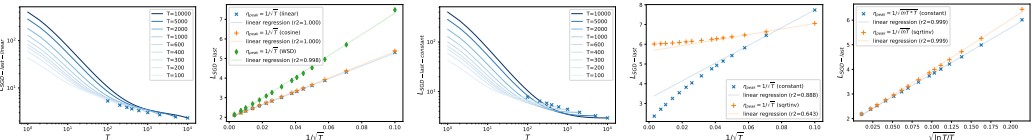

**Figure 1** Upper bound of SGD loss in Equation (2.4) with peak learning rate $\eta_{\text{peak}} = 1/\sqrt{T}$. Left-most is linear decaying schedule. Center is constant schedule.

# 3 GENERALIZATIONS TO DEEP LEARNING AND ADAPTIVE OPTIMIZERS

## 3.1 ABSTRACT FORM OF ANY-ITERATION LOSS

Our goal is to generalize the understanding **from convex analysis with SGD** beyond the scope of (2.4), **to non-convex deep learning and to general optimizers**. However, we cannot use (2.4) directly as we do not know $D, G, L_*$. Alternatively, we will estimate these quantities in a data-driven manner and work with an abstract form of the loss in (3.1).

**Generalization 1** (generalized from (2.4)). *For general optimizers and for deep learning, we have*

$$\mathbb{E}L(\mathbf{w}_\tau) \leq \tilde{L}_\infty + \frac{\tilde{D}^2}{2 \sum_{t=1}^\tau \eta_t} + \frac{\tilde{G}^2}{2} \left( \frac{\sum_{t=1}^\tau \eta_t^2}{\sum_{t=1}^\tau \eta_t} + \sum_{k=1}^{\tau-1} \frac{\eta_k}{\sum_{t=k+1}^\tau \eta_t} \frac{\sum_{t=k}^\tau \eta_t^2}{\sum_{t=k}^\tau \eta_t} \right) \tag{3.1}$$

In contrast to (2.4), we require the following modifications:

- The quantities $\tilde{D}$ and $\tilde{G}$ are no longer tied to $D = \|\mathbf{w}_0 - \mathbf{w}_*\|$ and $G^2 = \max_{\mathbf{w}} \mathbb{E}\|\mathbf{g}(\mathbf{w})\|\|^2$, i.e. we renounce the physical meaning of these quantities, because
  - general optimizers (e.g. momentum SGD, AdamW, Muon, parameter-efficient training such as LoRA) are not covered by (2.4) ever under the convex setting in Condition 2.1.
  - $D, G$ depend on the model architectures and datasets, which are hard to derive in practice.
- The irreducible loss is now $\tilde{L}_\infty = L(\lim_{\tau \to \infty} \mathbf{w}_\tau)$ instead of $L_* = L(\mathbf{w}_*)$. This accommodates the fact that there are infinite minima in deep learning, hence $\mathbf{w}_*$ is not unique and $L_*$ is not well-defined.

As we will observe in the following sections, (3.1) holds well at all iterations and becomes tight after some initial training. To be clear, we examine (3.1) on various

- models: ResNet (He et al., 2016), ViT (Dosovitskiy et al., 2020), GPT2 (Radford et al., 2019), vision-language models (which use LLAMA3 architecture (Dubey et al., 2024) as the language backbone).

- tasks: ImageNet (Deng et al., 2009), OpenWebText (Gokaslan et al., 2019), and Cauldron (Laurençon et al., 2024).
- optimizers: SGD, AdamW Loshchilov & Hutter (2019), Muon (Jordan et al., 2024), parameter-efficient training (LoRA (Hu et al., 2022)), etc.
- learning rate schedules: linear decay, cosine decay, WSD, constant, and square-root inverse.

## 3.2 GENERALIZING TO DEEP LEARNING

We start with SGD (same as (2.4)) but in the deep learning setting. In Figure 2, we train ResNet18 on ImageNet dataset with SGD under 4 learning rate schedules. The goodness of fit[1] for (3.1) is particularly evident for WSD and cyclic schedules: for WSD schedule, we see a sudden decrease of loss matching the decay of learning rate; for cyclic schedule, we see the periodic oscillation of loss matching that of learning rate. In particular, we use half the iterations to fit (solid line) and half to predict (dashed line). Our loss prediction by (3.1) is tight after a short period of training, with $R^2$ score $\geq 0.95$, highlighting the strong applicability of Generalization 1.

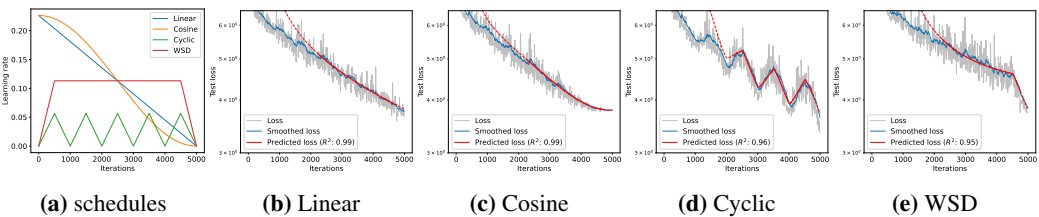

**(a)** schedules     **(b)** Linear     **(c)** Cosine     **(d)** Cyclic     **(e)** WSD

**Figure 2**   Sequence-to-sequence prediction by (3.1) for ResNet18 on ImageNet with SGD.

## 3.3 GENERALIZING TO ADAPTIVE OPTIMIZERS

We further test adaptive optimizers beyond SGD. Firstly, we compare ResNet18 trained with AdamW in Figure 3 to SGD in Figure 2 and observe the same patterns that (3.1) holds with $R^2$ score $\geq 0.95$.

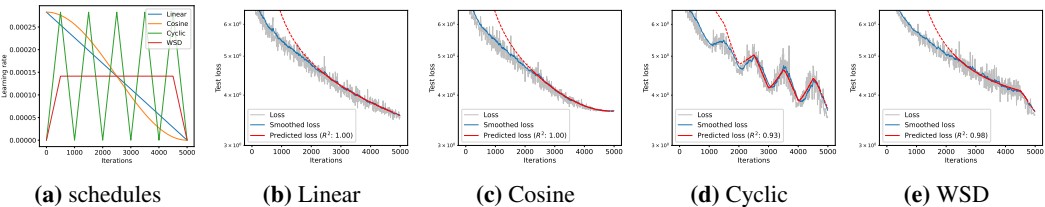

**(a)** schedules     **(b)** Linear     **(c)** Cosine     **(d)** Cyclic     **(e)** WSD

**Figure 3**   Sequence-to-sequence prediction by (3.1) for ResNet18 on ImageNet with AdamW.

We further pre-train GPT2 (124M) with AdamW and Muon-NSGD (Boreiko et al., 2025) optimizers, and observe consistent patterns in Figure 4 and Figure 5. The prediction $R^2$ scores are $\geq 0.95$ in all cases.

## 4 LOSS CHARACTERIZATION AT LAST ITERATION

Towards building a scaling law of loss and learning rate, we focus on the last iterate $\mathbf{w}_T$ and further abstract the loss characterization in (3.1).

**Generalization 2** (generalized from (2.5)). *For general optimizers under deep learning and for a qualified learning rate schedule with any peak learning rate $\eta_{\text{peak}}$, we have*

$$\mathbb{E}L(\mathbf{w}_T) \sim \tilde{L}_\infty + \frac{\tilde{q}_1^2}{T\eta_{\text{peak}}} + \eta_{\text{peak}}\tilde{q}_2^2 := L_{DL\text{-}last}(\eta_{\text{peak}}, T) \tag{4.1}$$

---

[1]We fit a non-negative linear regression $\boldsymbol{y} \sim \boldsymbol{X}\boldsymbol{\beta} + \tilde{L}_\infty$ where $\boldsymbol{y} \in \mathbb{R}^T, \boldsymbol{\beta} = [\tilde{D}, \tilde{G}]^\top, \boldsymbol{X} \in \mathbb{R}^{T\times 2}$, and specifically $y_\tau = L(\mathbf{w}_\tau), X_{\tau,1} = \frac{1}{2\sum_{t=1}^\tau \eta_t}, X_{\tau,2} = \frac{1}{2}\left(\frac{\sum_{t=1}^\tau \eta_t^2}{\sum_{t=1}^\tau \eta_t} + \sum_{k=1}^{\tau-1} \frac{\eta_k}{\sum_{t=k+1}^\tau \eta_t} \frac{\sum_{t=k}^\tau \eta_t^2}{\sum_{t=k}^\tau \eta_t}\right)$.

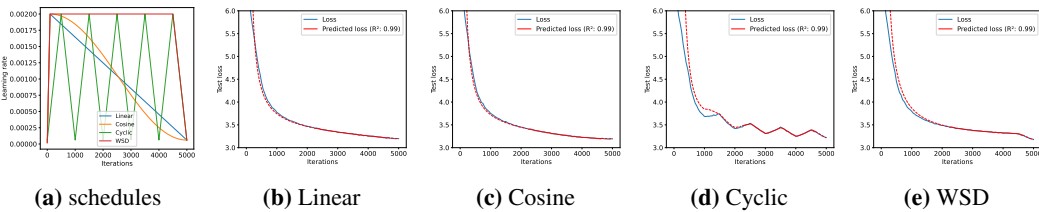

**(a)** schedules     **(b)** Linear     **(c)** Cosine     **(d)** Cyclic     **(e)** WSD

**Figure 4**   Sequence-to-sequence prediction by (3.1) for GPT2 on OpenWebText with AdamW.

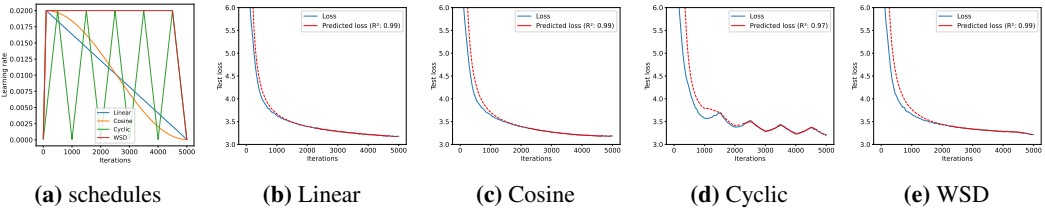

**(a)** schedules     **(b)** Linear     **(c)** Cosine     **(d)** Cyclic     **(e)** WSD

**Figure 5**   Sequence-to-sequence prediction by (3.1) for GPT2 on OpenWebText with Muon-NSGD.

In contrast to Generalization 1, we require the following modifications:

- We restrict our analysis to the qualified schedules that satisfy Condition 2.5, such as linear and cosine decay, instead of any schedules.
- We renounce the exact forms of $\tilde{q}_1$ and $\tilde{q}_2$ (e.g. $\tilde{q}_1 = \tilde{D}, \tilde{q}_2 = \tilde{G}$ for linear decay; $\tilde{q}_2 = \sqrt{1.061}\tilde{G}$ for cosine decay; $\tilde{q}_1 = \tilde{D}/(1+c)$ for WSD) and will derive these quantities in a data-driven manner.
- We focus on $\mathbf{w}_T$ rather than any iterate $\mathbf{w}_\tau$ and transform the inequality (3.1) to an approximate equality for moderately large $T$.

### 4.1 LOSS UNDER ANY PEAK LEARNING RATE

We test (4.1) using losses reported in (Li et al., 2025). These results were obtained under optimally tuned hyperparameters, where the language models (dense and Mixture-of-Experts; MoE) were trained on a mixture of web text, mathematics, and code. The training used AdamW optimizer with cosine decaying schedule (see details in Section 3.2 and Appendix A of (Li et al., 2025)).

We observe high $R^2$ scores ($\geq 0.95$) across different model sizes as well as model architectures, which empirically validates Generalization 2.

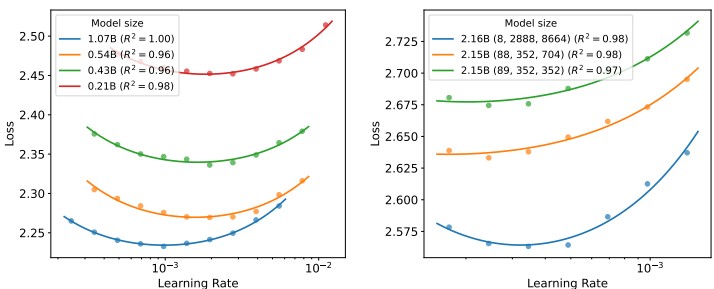

**Figure 6**   Loss versus learning rate for dense models (left) and MoE models (right) by (Li et al., 2025). In the right plot, the orange and green curves correspond to different architectures despite having the same model size. Specific model configurations are summarized in Appendix C.1.

### 4.2 LOSS UNDER OPTIMAL PEAK LEARNING RATE

We further derive and test (4.1) under the optimal $\eta_{\text{peak}}$ in Generalization 3.

**Generalization 3** (generalized from Corollary 2.4, part 2). *For general optimizers under deep learning and for a qualified learning rate schedule with optimal peak learning rate $\eta_{\text{peak}}$, we have* $\mathbb{E}L(\mathbf{w}_T) \sim \tilde{L}_\infty + 2\tilde{q}_1\tilde{q}_2/\sqrt{T}$.

To validate Generalization 3, we scrutinize the loss values and training horizons in (Hoffmann et al., 2022) for various models from $0.074B$ to $12.56B$, which were used to establish the compute-optimal scaling law for large language models. These runs trained Chinchilla models (same as Gopher (Rae et al., 2021)) with AdamW under cosine decaying schedule. The original loss values are not publicized in (Hoffmann et al., 2022) but they are reconstructed by (Besiroglu et al., 2024). Note that Hoffmann et al. (2022) only gives FLOPs[2], rather than the training iterations. Therefore, we translate the training horizon to token size=FLOPs/6/model size= batch size $\times T$.

**Table 2** Summary of linear regression on $\mathbb{E}L(\mathbf{w}_T)$ and $1/\sqrt{T} \cdot \text{batch size}$ from (Hoffmann et al., 2022; Besiroglu et al., 2024). Full table is deferred to Table 6. Runs for 2B model is visualized in Figure 7.

| model size(B) | num of horizons | $2\tilde{q}_1\tilde{q}_2$ | $\tilde{L}_\infty$ | $R^2$ score |
|---|---|---|---|---|
| 0.074 | 5 | 3.22e+04 | 2.825 | 0.991 |
| 0.140 | 7 | 3.04e+04 | 2.670 | 0.991 |
| 0.279 | 8 | 3.29e+04 | 2.498 | 0.999 |
| 0.425 | 8 | 3.27e+04 | 2.430 | 0.998 |
| 0.632 | 8 | 3.17e+04 | 2.367 | 0.998 |
| 1.143 | 10 | 3.10e+04 | 2.275 | 0.998 |
| 1.429 | 9 | 3.18e+04 | 2.253 | 0.996 |
| 1.611 | 9 | 3.36e+04 | 2.228 | 0.995 |
| 2.004 | 8 | 3.62e+04 | 2.178 | 0.999 |
| 2.280 | 7 | 4.41e+04 | 2.128 | 1.000 |
| 2.979 | 10 | 5.90e+04 | 2.016 | 0.990 |
| 4.519 | 6 | 3.83e+04 | 2.106 | 0.978 |
| 6.792 | 8 | 4.66e+04 | 2.023 | 0.999 |
| 9.290 | 4 | 4.29e+04 | 2.046 | 0.988 |
| 12.56 | 3 | 4.23e+04 | 2.053 | 1.000 |

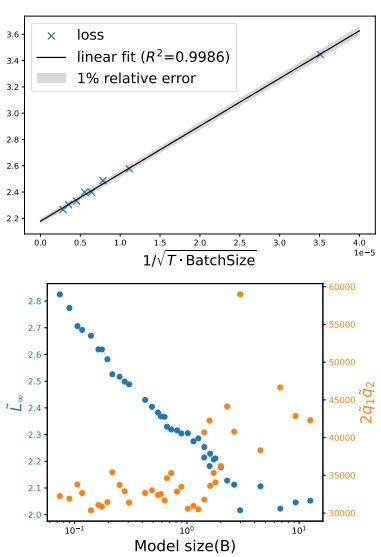

**Figure 7** Upper: 2B model precisely fits Generalization 3 across a wide range of training horizons. Lower: As model size increases, $\tilde{L}_\infty$ log-linearly decreases and $2\tilde{q}_1\tilde{q}_2$ roughly increases.

In Table 2, we consistently observe precise fitting of Generalization 3 with $R^2 \geq 0.978$, up to FLOPs=1e22 and over 300B tokens. We note that all loss values are within 1% relative error of our prediction in Figure 7.

## 5  TWO-DIMENSIONAL SCALING LAW FOR LEARNING RATE

The success of transferring insights from convex analysis to deep learning underscores the appeal of predicting and controlling the loss via learning rate. In this section, we propose two-dimensional scaling law of optimal loss and learning rate at various model sizes $(N)$ and training horizons $(T)$.

Notably, we must take one step further from Generalization 3, because we do not know the optimal $\eta_{\text{peak}}$, except it has a form $\frac{\tilde{q}_1}{\tilde{q}_2\sqrt{T}}$ with unknown $\tilde{q}_1$ and $\tilde{q}_2$.

**Generalization 4** (generalized from Corollary 2.4, part 1). *For general optimizers under deep learning and for a qualified learning rate schedule with scaled peak learning rate $\eta_{\text{peak}} = \eta_{\text{ref}}/\sqrt{T}$, we have*

$$\mathbb{E}L(\mathbf{w}_T) \sim \tilde{L}_\infty + \tilde{Q}(\eta_{\text{ref}})/\sqrt{T}, \forall \eta_{\text{ref}}.$$

---

[2]FLOP$\approx 6 \times$ token size $\times$ model size $= 6 \times$ batch size $\times$ iterations $\times$ model size. The batch size is not released in this work and assumed to be constant across all runs.

In words, we have $O(1/\sqrt{T})$ loss convergence under $1/\sqrt{T}$-scaled $\eta_{\text{peak}}$, including but not limited to the optimal $\eta_{\text{ref}}$ in Generalization 3. Notice this expression is linear in $1/\sqrt{T}$ and can be visualized by a straight line. We seek the optimal $\eta_{\text{ref}}$ by multiple small-scale runs and we estimate $\tilde{L}_\infty$ and $\tilde{Q}$ via linear regression in Section 5.2 and Section 5.3.

Finally, we present a scaling law that predicts both loss and optimal learning rate:

$$\mathbb{E}L(N,T) \sim \tilde{L}_\infty(\eta_{\text{ref}}^*; N) + \tilde{Q}(\eta_{\text{ref}}^*; N)/\sqrt{T}$$
$$\eta_{\text{peak}}^*(N,T) \sim \eta_{\text{peak}}^*(N_{\text{small}}, T_{\text{small}})/\sqrt{T/T_{\text{small}}}$$
(5.1)

where $N_{\text{small}}$ is a smaller model with $N_{\text{small}} \leq N$, and $T_{\text{small}}$ is a shorter training horizon with $T_{\text{small}} \leq T$, so that we can use small-scale training to inform the large-scale training's hyperparameter.

## 5.1 EXPERIMENT SETTINGS

We train GPT2 language models on OpenWebText for various training horizons, following nanoGPT codebase (Karpathy, 2023). We apply two optimizers: AdamW and Muon-NSGD, both with 0.01 weight decay and without gradient clipping. Here Muon-NSGD is adapted from the original Muon by (1) optimizing all 2D tensors with Muon and other tensors with normalized SGD, i.e. NSGD, and (2) using a single learning rate for Muon and NSGD. We use the cosine decaying learning rate schedule that decays to 0 with 2% warm-up. We conduct 240 runs, which are 4 settings (0.1B/AdamW, 0.1B/Muon-NSGD, 1B/Muon-NSGD, and 7B/Muon-NSGD), 10 training horizons from 100 to 500k iterations, and 6 $\eta_{\text{ref}}$ from 0.01 to 30.0). See more details in Appendix B.

To illustrate the generality of our approach, we conduct *additional experiments* in Appendix D for parameter-efficient training (LoRA (Hu et al., 2022)) and ablations over key hyperparameters, such as weight decay, gradient clipping, momentum coefficient, batch size, and random seeds. We consistently observe $O(1/\sqrt{T})$ loss convergence with scaled learning rate across different regimes.

## 5.2 SCALING ACROSS TRAINING HORIZONS

We train GPT2 (0.1B) for $T$ ranging from 100 to 500k steps. We obtain good fitting of linear regression over $1/\sqrt{T} < 0.02$, i.e. $T > 2.5k$ which indicates that Generalization 4 becomes predictive very early during training. This observation is consistent for two optimizers – AdamW in Figure 8 and Muon-NSGD in Figure 9.

From the intercepts of fitted lines, we determine leverage the optimal $\eta_{\text{peak}}^* = \eta_{\text{ref}}^*/\sqrt{T}$ where $\eta_{\text{ref}}^* = 10$ for Muon-NSGD and $\eta_{\text{ref}}^* = 0.3$ for AdamW. As we observe from the zoom-in plots in Figure 9 and later in Figure 10, the extrapolation of our scaling law across training horizons is $\approx 80\times$.

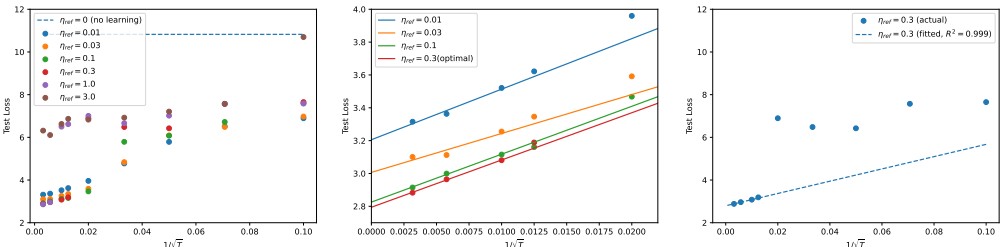

**Figure 8**  Loss values (dots) and $1/\sqrt{T}$ prediction for GPT2 (0.1B) with AdamW.

## 5.3 SCALING ACROSS TRAINING HORIZONS AND MODEL SIZES

We further train GPT2 1B and 7B models with Muon-NSGD, using the same $\eta_{\text{ref}}^*$ that we transfer from GPT 0.1B model by (5.1). Taking a closer look at 7B model losses, we observe precise fitting of our scaling law, extrapolating across model sizes by $70\times$.

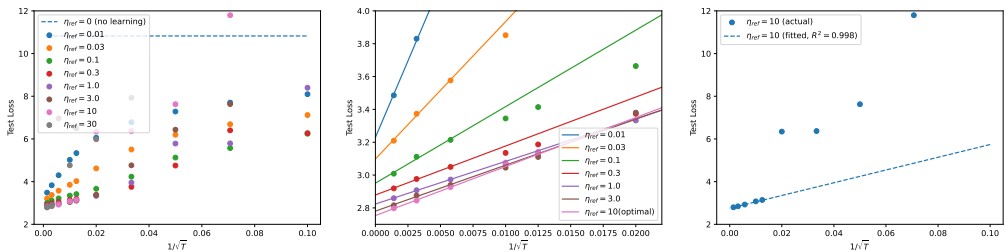

**Figure 9** Loss values (dots) and $1/\sqrt{T}$ prediction for GPT2 (0.1B) with Muon-NSGD.

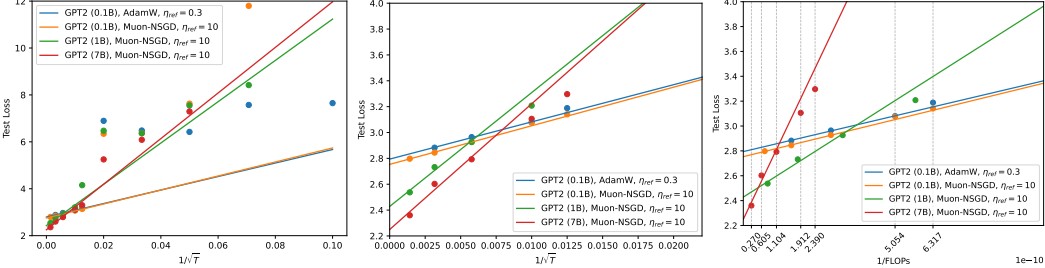

**Figure 10** Loss values (dots) and $1/\sqrt{T}$ or $1/\sqrt{\text{FLOPs}}$ prediction for 4 settings.

## 5.4 SCALING ON MULTI-MODAL MODELS

We finetune vision-language models (VLM) with $\approx$1B parameters on the Cauldron dataset (Laurençon et al., 2024), following nanoVLM codebase (Wiedmann et al., 2025). We apply AdamW optimizer with cosine decaying learning rate and 3% warmup. We test three values of $\eta_{\text{ref}}$ across training horizons up to 14.4k iterations. In Figure 11, we observe good fitting when $T > 2000$, thus generalizing our scaling law to multi-modal models. We note that VLM has multiple components (language backbone, vision backbone, and modality projector) and each has a separate learning rate, all of which are $1/\sqrt{T}$ scaled.

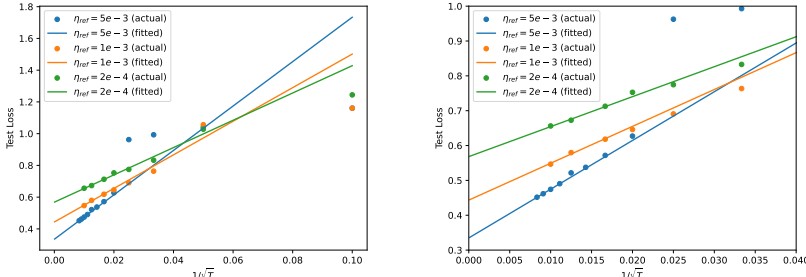

**Figure 11** Loss values (dots) and $1/\sqrt{T}$ prediction for multi-modal VLM with AdamW. Right plot is zoomed.

## 6 CONCLUSION

This paper presents a path, starting from a rigorous analysis that is restricted to SGD and convex loss, and generalizing to non-convex deep learning with general optimizers beyond the support of theory. Along the path, we heavily rely on data-driven method to fit our prediction of loss and learning rate. Our findings support (I) convex-like behavior in deep learning, which can be characterized by sequence-to-sequence prediction in (2); (II) asymptotic prediction of $O(1/\sqrt{T})$ loss convergence and $O(1/\sqrt{T})$ learning rate in deep learning; (III) a scaling law that extrapolates across training horizons and model sizes. We note some limitations of this paper, including the fact that our approach fails to predict test loss but continues to predict training loss when overfitting is severe (see Figure 12), and the lack of understanding why convex-like behaviors exist in various architectures and how many iterations it takes for deep learning to be characterizable by such behaviors.

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

# A  PROOFS

## A.1  DERIVATION OF TABLE 1

**Theorem 1.** *For SGD under Condition 2.1, Equation (2.4) gives:*

***Case 1: Constant learning rate** ($\eta_t = \eta_{\text{peak}}$ for all t):*

$$\mathbb{E}L(\mathbf{w}_T) \lesssim L_* + \frac{D^2}{2T\eta_{\text{peak}}} + \frac{\eta_{\text{peak}}G^2}{2}\ln T := L_{\textit{SGD-last-constant}}(\eta_{\text{peak}}; T)$$

*The upper bound is minimized at:*

$$\min_{\eta} L_{\textit{SGD-last-constant}}(\eta_{\text{peak}}; T) = L_* + DG\sqrt{\frac{\ln T}{T}}, \quad \text{with } \eta^*_{\text{peak}}(T) = \frac{D}{G\sqrt{\ln T \cdot T}}.$$
(A.1)

***Case 2: Square-root inverse learning rate** ($\eta_t = \eta_{\text{peak}}/\sqrt{t}$):*

$$\mathbb{E}L(\mathbf{w}_T) \lesssim L_* + \frac{D^2}{4\sqrt{T}\eta_{\text{peak}}} + \frac{\eta_{\text{peak}}G^2\ln T}{4\sqrt{T}} := L_{\textit{SGD-last-sqrt-inv}}(\eta_{\text{peak}}; T)$$

*The upper bound is minimized at:*

$$\min_{\eta_{\text{peak}}} L_{\textit{SGD-last-sqrt-inv}}(\eta_{\text{peak}}; T) = L_* + DG\sqrt{\frac{\ln T}{4T}}, \quad \text{with } \eta^*_{\text{peak}}(T) = \frac{D}{G\sqrt{\ln T}}.$$
(A.2)

***Case 3: Linearly decaying learning rate** ($\eta_t = \eta_{\text{peak}}(1 - t/T)$):*

$$\mathbb{E}L(\mathbf{w}_T) \lesssim L_* + \frac{D^2}{T\eta_{\text{peak}}} + \eta_{\text{peak}}G^2 := L_{\textit{SGD-last-linear}}(\eta_{\text{peak}}; T)$$

*The upper bound is minimized at:*

$$\min_{\eta_{\text{peak}}} L_{\textit{SGD-last-linear}}(\eta_{\text{peak}}; T) = L_* + 2DG\sqrt{\frac{1}{T}}, \quad \text{with } \eta^*_{\text{peak}}(T) = \frac{D}{G\sqrt{T}}.$$
(A.3)

***Case 4: Cosine decaying learning rate** ($\eta_t = \eta_{\text{peak}}\frac{1+\cos(\pi t/T)}{2}$):*

$$\mathbb{E}L(\mathbf{w}_T) \lesssim L_* + \frac{D^2}{T\eta_{\text{peak}}} + \eta_{\text{peak}}G^2 \cdot 1.061 := L_{\textit{SGD-last-cosine}}(\eta_{\text{peak}}; T)$$

*The upper bound is minimized at:*

$$\min_{\eta_{\text{peak}}} L_{\textit{SGD-last-cosine}}(\eta_{\text{peak}}; T) = L_* + 2DG\sqrt{\frac{1.061}{T}}, \quad \text{with } \eta^*_{\text{peak}}(T) = \frac{D}{G\sqrt{1.061T}}.$$
(A.4)

***Case 5: Warmup-stable-decay learning rate** ( $\eta_t = \begin{cases} \eta_{\text{peak}} & \text{if } t < cT \\ \eta_{\text{peak}}\frac{T-t}{T-cT} & \text{if } t \geq cT \end{cases}$):*

$$\mathbb{E}L(\mathbf{w}_T) \lesssim L_* + \frac{D^2}{(1+c)\eta_{\text{peak}}T} + \eta_{\text{peak}}G^2\left[1 + \frac{1}{2}\ln\left(\frac{1+c}{1-c}\right)\right] := L_{\textit{SGD-last-wsd}}(\eta_{\text{peak}}; T).$$

*The upper bound is minimized at:*

$$\min_{\eta_{\text{peak}}} L_{\textit{SGD-last-wsd}}(\eta_{\text{peak}}; T) = L_* + 2DG\sqrt{\frac{1 + \frac{1}{2}\ln\left(\frac{1+c}{1-c}\right)}{(1+c)T}},$$
(A.5)

$$\text{with} \quad \eta^*_{\text{peak}}(T) = \frac{D}{G}\sqrt{(1+c)\left(1 + \frac{1}{2}\ln\left(\frac{1+c}{1-c}\right)\right)T}.$$
(A.6)

*Proof of Theorem 1.* For the ease of presentation, we work in the continuous regime and translate (2.3) to

$$L_{\text{SGD-any}}(\tau) = L_* + \frac{D^2}{2\int_0^\tau \eta_t dt} + \frac{\int_0^\tau \eta_t^2 dt}{2\int_0^\tau \eta_t dt} G^2 + \frac{G^2}{2}\int_0^{\tau-1} \frac{\eta_k \left(\int_{k-1}^\tau \eta_t^2 dt\right)}{\int_k^\tau \eta_t dt \int_{k-1}^\tau \eta_t dt} dk.$$

We will later show Condition 2.5 is also the continuous version of (2.3).

**Constant learning rate**   $\eta_t = \eta$. Hence, $\int_k^T \eta_t dt = (T-k)\eta$, $\int_k^T \eta_t^2 dt = (T-k)\eta^2$. Then

$$
\begin{aligned}
L_{\text{SGD-last-constant}}(T) - L_* &= \frac{D^2}{2T\eta} + \frac{G^2\eta}{2} + \frac{G^2}{2}\int_0^{T-1} \frac{\eta_k \left(\int_{k-1}^T \eta_t^2 dt\right)}{\int_k^T \eta_t dt \int_{k-1}^T \eta_t dt} dk \\
&= \frac{D^2}{2T\eta} + \frac{G^2\eta}{2} + \frac{G^2}{2}\int_0^{T-1} \frac{\eta}{T-k} dk \\
&= \frac{D^2}{2T\eta} + \frac{G^2}{2}\eta \ln T.
\end{aligned}
$$

**Square-root inverse learning rate**   $\eta_t = \eta_{\text{peak}}/\sqrt{t+1}$, then $\int_k^T = 2\eta(\sqrt{T+1} - \sqrt{k+1})$ and $\int_k^T \eta_t^2 = \eta^2 \ln\left(\frac{T+1}{k+1}\right)$, and we can write

$$
\begin{aligned}
L_{\text{SGD-last-sqrt-inv}}(T) - L_* =& \frac{D^2}{4\eta(\sqrt{T+1} - 1)} + \frac{\eta \ln(T+1)}{4(\sqrt{T+1} - 1)} G^2 \\
&+ \frac{\eta G^2}{8}\int_0^{T-1} \frac{\ln\left(\frac{T+1}{k}\right)}{\sqrt{k+1}(\sqrt{T+1}) - \sqrt{k+1}(\sqrt{T+1} - \sqrt{k})} dk.
\end{aligned}
$$

Let $A = \sqrt{T+1}$ and with the change of variable $u = A - \sqrt{k}$, $du = -\frac{1}{2(A-u)} dk$ we can write $\sqrt{T+1} - \sqrt{k} = u$, $\sqrt{T+1} - \sqrt{k+1} = u + O(\frac{1}{A})$, $\sqrt{k+1} = A - u + O(\frac{1}{A})$. Therefore,

$$\frac{1}{\sqrt{k+1}(\sqrt{T+1}) - \sqrt{k+1}(\sqrt{T+1} - \sqrt{k})} \approx \frac{1}{(A-u)u^2}$$

and

$$\ln\left(\frac{T+1}{k}\right) = -2\ln\left(1 - \frac{u}{A}\right).$$

Also notice the upper limit $k = T - 1$ corresponds to $u \approx \frac{1}{2A}$ and the lower limit is $u = A$, so the integral can be approximated by

$$I(T) = \int_0^{T-1} \frac{\ln\left(\frac{T+1}{k}\right)}{\sqrt{k+1}(\sqrt{T+1}) - \sqrt{k+1}(\sqrt{T+1} - \sqrt{k})} dk \approx -4\int_{1/(2A)}^A \frac{\ln(1 - u/A)}{u^2} du.$$

This integral can be calculated exactly with a change of variable $\epsilon = \frac{1}{2A^2}$ as

$$-\frac{4}{A}\left(\frac{\ln(1-\epsilon)}{\epsilon} + \ln\epsilon - \ln(1-\epsilon)\right) = -\frac{4}{A}(\ln\epsilon - 1 + o(1)) \approx \frac{4\ln(T+1) + 6.7726}{\sqrt{T+1}} + o(1/\sqrt{T}).$$

Combining the above we have

$$
\begin{aligned}
&L_{\text{SGD-last-sqrt-inv}}(T) - L_* \\
&\approx \frac{D^2}{4\eta(\sqrt{T+1} - 1)} + \frac{\eta \ln(T+1)}{4(\sqrt{T+1} - 1)} G^2 + \frac{4\ln(T+1) + 6.7726}{\sqrt{T+1}} + o(1/\sqrt{T}) \\
&\sim \frac{D^2}{4\eta\sqrt{T}} + \frac{\eta \ln(T)}{4\sqrt{T}} G^2 + O\left(\frac{\ln(T)}{\sqrt{T}}\right).
\end{aligned}
$$

**Linear decay learning rate**   $\eta_t = \eta(1 - t/T)$ and it is not hard to get $\int_k^T \eta_t dt = \eta(T-k)^2/2T$, $\int_k^T \eta_t^2 dt = \eta^2(T-k)^3/3T^2$.

We can write down

$$
\begin{aligned}
L_{\text{SGD-last-linear}}(T) - L_* &= \frac{D^2}{\eta T} + \frac{\eta^2 T/3}{\eta T}G^2 + \frac{G^2}{2}\int_0^{T-1} \frac{\eta_k \eta^2(T-k+1)^3/3T^2}{\eta(T-k)^2/2T * (T-k+1)^2/2T}dk \\
&= \frac{D^2}{\eta T} + \frac{\eta G^2}{3} + \frac{2\eta G^2}{3T}\int_0^{T-1}(1 + \frac{1}{(T-k)})dk \\
&= \frac{D^2}{\eta T} + \frac{\eta G^2}{3} + \frac{2\eta G^2}{3T}(T - 1 + \ln T) \\
&= \frac{D^2}{\eta T} + \frac{\eta G^2}{3}(3 + 2\ln T/T - 2/T).
\end{aligned}
$$

**Cosine decay learning rate**   $\eta_t = \frac{1}{2}(1 + \cos(\pi t/T))\eta$.

We derive

$$
\begin{aligned}
A(k) := \int_k^T \eta_t dt &= \eta \int_k^T \frac{1 + \cos\left(\frac{\pi t}{T}\right)}{2}dt \\
&= \eta\left(\frac{1}{2}\left[(T-k)\right] + \frac{1}{2}\int_k^T \cos\left(\frac{\pi t}{T}\right)dt\right) \\
&= \eta\left(\frac{T-k}{2} + \frac{1}{2}\cdot\frac{T}{\pi}\left[\sin\left(\frac{\pi t}{T}\right)\right]_{t=k}^{t=T}\right) \\
&= \eta\left(\frac{T-k}{2} - \frac{T}{2\pi}\sin\left(\frac{\pi k}{T}\right)\right).
\end{aligned}
$$

and

$$
\begin{aligned}
B(k) := \int_k^T \eta_t^2 dt &= \eta^2 \int_k^T \left(\frac{3}{8} + \frac{1}{2}\cos\frac{\pi t}{T} + \frac{1}{8}\cos\frac{2\pi t}{T}\right)dt \\
&= \eta^2\left(\frac{3(T-k)}{8} + \frac{1}{2}\cdot\frac{T}{\pi}\left[\sin\frac{\pi t}{T}\right]_k^T + \frac{1}{8}\cdot\frac{T}{2\pi}\left[\sin\frac{2\pi t}{T}\right]_k^T\right) \\
&= \eta^2\left(\frac{3(T-k)}{8} - \frac{T}{2\pi}\sin\left(\frac{\pi k}{T}\right) - \frac{T}{16\pi}\sin\left(\frac{2\pi k}{T}\right)\right).
\end{aligned}
$$

Finally, substituting them into main expression gives

$$
\begin{aligned}
&\frac{D^2}{2A(0)} + \frac{B(0)}{2A(0)}G^2 + \frac{G^2}{2}\int_0^{T-1}\frac{\eta_k B(k-1)}{A(k)A(k-1)}dk, \\
&= \frac{D^2}{T\eta} + \frac{3\eta G^2}{8} + \frac{\eta G^2}{2}\int_0^{T-1}\frac{\frac{1}{2}\left(1+\cos(\frac{\pi k}{T})\right)B(k-1)}{A(k)A(k-1)}dk \\
&= \frac{D^2}{T\eta} + \frac{3\eta G^2}{8} + \frac{\eta G^2}{4}\int_0^{T-1}\frac{\left(1+\cos(\frac{\pi k}{T})\right)\left[\frac{3(T-k+1)}{8} - \frac{T}{2\pi}\sin\left(\frac{\pi(k-1)}{T}\right) - \frac{T}{16\pi}\sin\left(\frac{2\pi(k-1)}{T}\right)\right]}{\left(\frac{T-k}{2} - \frac{T}{2\pi}\sin(\frac{\pi k}{T})\right)\left(\frac{T-k+1}{2} - \frac{T}{2\pi}\sin(\frac{\pi(k-1)}{T})\right)}dk.
\end{aligned}
$$

By a change of variable $x = \frac{k}{T}$, the last integral behaves like

$$
\begin{aligned}
&T\int_0^{1-1/T}\left(\frac{(1+\cos(\pi x))B(Tx)}{A^2(Tx)} + O\left(\frac{1}{T^2}\right)\right)dx \\
&= \int_0^1 \frac{(1+\cos\pi x)\left[\frac{3(1-x)}{8} - \frac{1}{2\pi}\sin(\pi x) - \frac{1}{16\pi}\sin(2\pi x)\right]}{\left(\frac{1-x}{2} - \frac{1}{2\pi}\sin(\pi x)\right)^2}dx + O\left(\frac{1}{T}\right) \\
&\approx 2.7443 + O\left(\frac{1}{T}\right).
\end{aligned}
$$

Therefore,

$$L_{\text{SGD-last-cosine}}(T) - L_* \approx \frac{D^2}{T\eta} + \frac{3\eta G^2}{8} + \frac{\eta G^2}{4} \times (2.7443 + O(\frac{1}{T})) \approx \frac{D^2}{T\eta} + \eta G^2(1.061 + O(\frac{1}{T})).$$

**Warmup-stable-decay learning rate** $\quad \eta_t = \begin{cases} \eta & \text{if } t < cT \\ \eta\frac{T-t}{T-cT} & \text{if } t \geq cT \end{cases}$. We can get

$$\int_k^T \eta_t dt = \begin{cases} \eta(cT - k) + \frac{(1-c)\eta T}{2} & \text{if } k < cT, \\ \frac{\eta(T-k)^2}{2(T-cT)} & \text{if } k \geq cT, \end{cases}$$

and

$$\int_k^T \eta_t^2 dt = \begin{cases} \eta^2(cT - k) + \frac{\eta^2}{3}(T - cT) & \text{if } k < cT, \\ \frac{\eta^2}{3(T-cT)^2}(T - k)^3 & \text{if } k \geq cT. \end{cases}$$

We can write

$$
\begin{aligned}
&L_{\text{SGD-last-WSD}}(T) - L_* \\
&= \frac{D^2}{(1+c)\eta T} + \frac{\frac{1+2c}{3}\eta^2 T}{(1+c)\eta T}G^2 \\
&\quad + \frac{G^2}{2}\left[\int_0^{cT} \frac{\eta_k\left(\int_{k-1}^T \eta_t^2 dt\right)}{\int_k^T \eta_t dt \int_{k-1}^T \eta_t dt}dk + \int_{cT}^{T-1} \frac{\eta_k\left(\int_{k-1}^T \eta_t^2 dt\right)}{\int_k^T \eta_t dt \int_{k-1}^T \eta_t dt}dk\right] \\
&\approx \frac{D^2}{(1+c)\eta T} + \frac{\frac{1+2c}{3}\eta^2 T}{(1+c)\eta T}G^2 \\
&\quad + \frac{G^2}{2}\left[\int_0^{cT} \frac{\eta\left(\eta^2(cT - k) + \frac{\eta^2}{3}(T - cT)\right)}{(\eta(cT - k) + \frac{(1-c)\eta T}{2})^2}dk + \int_{cT}^{T-1} \frac{\eta\frac{T-t}{T-cT}\left(\frac{\eta^2}{3(T-cT)^2}(T - k)^3\right)}{(\frac{\eta(T-k)^2}{2(T-cT)})^2}dk\right] \\
&= \frac{D^2}{(1+c)\eta T} + \frac{(1+2c)\eta}{3(1+c)}G^2 + \frac{\eta G^2}{2}\left[\ln\left(\frac{1+c}{1-c}\right) - \frac{2c}{3(1+c)} + \frac{4}{3}\left(1 - \frac{1}{(1-c)T}\right)\right] \\
&\sim \frac{D^2}{(1+c)\eta T} + \frac{(1+2c)\eta}{3(1+c)}G^2 + \frac{\eta G^2}{2}\left[\ln\left(\frac{1+c}{1-c}\right) - \frac{2c}{3(1+c)} + \frac{4}{3}\right] \\
&= \frac{D^2}{(1+c)\eta T} + \eta G^2\left[1 + \frac{1}{2}\ln\left(\frac{1+c}{1-c}\right)\right]
\end{aligned}
$$

$\square$

## A.2 DERIVATION OF CONDITION 2.5

Recall the loss bound at last iterate in (2.4) is

$$L_* + \frac{D^2}{2\sum_{t=1}^T \eta_t} + \frac{G^2}{2}\left(\frac{\sum_{t=1}^T \eta_t^2}{\sum_{t=1}^T \eta_t} + \sum_{k=1}^{T-1} \frac{\eta_k}{\sum_{t=k+1}^T \eta_t} \frac{\sum_{t=k}^T \eta_t^2}{\sum_{t=k}^T \eta_t}\right)$$

We can rewrite the last term by applying Theorem 2, with $x_t = \eta_t^2$.

**Theorem 2.** *For any sequence $\{x_t\}$, if $x_T/\eta_T = 0$, then*

$$\frac{1}{\sum_{t=1}^T \eta_t}\sum_{t=1}^T x_t + \sum_{k=1}^{T-1} \frac{\eta_k}{\sum_{t=k+1}^T \eta_t \sum_{t=k}^T \eta_t}\sum_{t=k}^T x_t = \sum_{t=1}^{T-1}\left(\frac{x_t}{\sum_{k=t+1}^T \eta_k}\right)$$

*Proof of Theorem 2.* We firstly simplify the notation by denoting $A_k = \frac{\eta_k}{\sum_{t=k+1}^T \eta_t \sum_{t=k}^T \eta_t} = \frac{1}{\sum_{t=k+1}^T \eta_t} - \frac{1}{\sum_{t=k}^T \eta_t}$ for $k < T$.

With this notation, we write the left hand side of Theorem 2 as

$$\frac{1}{\sum_{t=1}^{T}\eta_t}\sum_{t=1}^{T}x_t + \sum_{k=1}^{T-1}\left(A_k\sum_{t=k}^{T}x_t\right) = \frac{1}{\sum_{t=1}^{T}\eta_t}\sum_{t=1}^{T}x_t + \sum_{k=1}^{T-1}\left(A_k\sum_{t=k}^{T-1}x_t\right) + \sum_{k=1}^{T-1}A_k x_T$$

Now we can exchange the double sum by noticing that

$$\sum_{k=1}^{T-1}\left(A_k\sum_{t=k}^{T-1}x_t\right) = \sum_{t=1}^{T-1}\left(x_t\sum_{k=1}^{t}A_k\right) = \sum_{1\le k\le t\le T-1}A_k x_t$$

Therefore the left hand side of Theorem 2 becomes

$$\frac{1}{\sum_{t=1}^{T}\eta_t}\sum_{t=1}^{T}x_t + \sum_{t=1}^{T-1}\left(x_t\sum_{k=1}^{t}A_k\right) + x_T\left(\sum_{k=1}^{T-1}A_k\right)$$

Splitting the first term, we get

$$\sum_{t=1}^{T-1}x_t\left(\frac{1}{\sum_{t=1}^{T}\eta_t}+\sum_{k=1}^{t}A_k\right) + x_T\left(\frac{1}{\sum_{t=1}^{T}\eta_t}+\sum_{k=1}^{T-1}A_k\right) \tag{A.7}$$

in which the term in big bracket can be greatly simplified by telescoping the sum,

$$\frac{1}{\sum_{t=1}^{T}\eta_t}+\sum_{k=1}^{t}A_k = \frac{1}{\sum_{s=1}^{T}\eta_s}+\sum_{k=1}^{t}\left(\frac{1}{\sum_{s=k+1}^{T}\eta_s}-\frac{1}{\sum_{s=k}^{T}\eta_s}\right) = \frac{1}{\sum_{s=t+1}^{T}\eta_s}.$$

In particular, with $t=T-1$, this is just $1/\eta_T$.

Substituting back to right hand side of (A.7), we finally obtain an equivalent form of the left hand side of Theorem 2 as follows, after using $x_T/\eta_T = 0$:

$$\sum_{t=1}^{T-1}x_t\left(\frac{1}{\sum_{s=t+1}^{T}\eta_s}\right) + \frac{x_T}{\eta_T}$$

$\square$

Now we have

$$L_* + \frac{D^2}{2\sum_{t=1}^{T}\eta_t} + \frac{G^2}{2}\sum_{t=1}^{T-1}\left(\frac{\eta_t^2}{\sum_{k=t+1}^{T}\eta_k}\right) \tag{A.8}$$

However, (A.8) has some singularity in the last term at $t=T$, where the denominator may be zero (if learning rate decays) and not divisible. To work around, we replace the discrete summation by continuous integral to get

$$L_* + \frac{D^2}{2\int_0^T \eta_t dt} + \frac{G^2}{2}\int_0^T\left(\frac{\eta_t^2}{\int_t^T \eta_k dk}\right)dt$$

### A.3 SUMMARY OF GENERALIZATIONS

**Table 3** Summary of generalizations in this work and coefficients to be fitted in a data-driven way.

|  | related experiments | coefficients to fit |
|---|---|---|
| Generalization 1 | Figure 2-5 | $\tilde{L}_\infty, \tilde{D}, \tilde{G}$ |
| Generalization 2 | Figure 6 | $\tilde{L}_\infty, \tilde{q}_1, \tilde{q}_2$ |
| Generalization 3 | Figure 7, Table 2 | $\tilde{L}_\infty, \tilde{q}_1\tilde{q}_2$ |
| Generalization 4 | Figure 8-11 | $\tilde{L}_\infty, \tilde{Q}(\eta_{\text{ref}})$ |

## B  MORE EXPERIMENT SETTINGS

**Figure 1:**  WSD uses 10% warmup and 10% decay.

**Figure 2:**  Optimizer is SGD without momentum or weight decay, batch size 256.

Linear decays from peak learning rate 0.225 to 0. Cosine uses peak learning rate 0.225. Cyclic uses 5 triangular cycles and peak learning rate is 0.056. WSD uses 10% warmup and 10% decay, peak learning rate 0.113.

**Figure 3**  Optimizer is AdamW: $\beta_1 = 0.9, \beta_2 = 0.999$, weight decay= 0.01, batch size 256.

Linear decays from peak learning rate 0.0003 to 0. Cosine uses peak learning rate 0.0003. Cyclic uses 5 triangular cycles and peak learning rate is 0.0003. WSD uses 10% warmup and 10% decay, peak learning rate 0.00014.

**Figure 4 and Figure 5**  We use two optimizers: AdamW and Muon-NSGD. Here Muon-NSGD uses Muon and NSGD with the same learning rate as in (Boreiko et al., 2025): denoting $\mathbf{W}$ as a layer's parameter and ignoring weight decay for brevity,

$$\text{Muon:} \mathbf{W}_{t+1} = \mathbf{W}_t - \eta \cdot \text{NS}(\mathbf{m}_t)$$
$$\text{NSGD:} \mathbf{W}_{t+1} = \mathbf{W}_t - \eta \cdot \mathbf{m}_t / \|\mathbf{m}_t\|_2$$

where NS is the Newton-Schulz matrix iteration and $\mathbf{m}_t$ is the momentum.

Each optimizer is trained under four learning rate schedulers, i.e., linear decay with warm-up, cosine decay with warm-up, cyclic and WSD. For schedulers with warm-up, we allocate 2% of the total steps (i.e., 100 of 5000 steps) to warmup. For WSD, we start to decay the learning rate from the last 10% of the steps. For AdamW, the peak learning rate is 2e-3. For Muon-NSGD, the peak learning rate is 2e-2.

**Section 5**  All runs are trained with bf16 mixed-precision training. AdamW uses $\beta_1 = 0.9, \beta_2 = 0.95$. Muon-NSGD uses momentum 0.95. 0.1B models use batch size 512. 1B/7B models use batch size 64. We train with 1024 sequence length.

VLM models use SmolLM2-360M-Instruct (Allal et al., 2025) as the language model and siglip2-base-patch16-512 (Tschannen et al., 2025) as the vision model. These models use different learning rates instead of a single one. For example, when $\eta_{\text{ref}} = 5e - 3$, the vision and language backbones use $\eta_{\text{ref}}/\sqrt{T}$ as the learning rate while the modality projector uses $100\eta_{\text{ref}}/\sqrt{T}$. As a consequence, all learning rates are $1/\sqrt{T}$ scaled. AdamW uses default hyperparameters with batch size being 64. Sequence length 8192, max image size 2048, max images per example 8, max images per knapsack 36.

## C  DETAILS OF SECTION 4

### C.1  FIGURE 6

The model configurations of Figure 6 can be found in Table 4 and Table 5. All other parameters not specified here follow the default setting in (Li et al., 2025).

**Table 4**  Dense model configurations. Model size is measured in billions of parameters (B). In columns, $d_{model}$ = hidden dimension, $d_{ff}$ = feed-forward width, $n_{heads}$ = number of attention heads, $n_{layers}$ = number of layers, $B$ = batch size, $T$ = total iterations.

| Model Size (B) | $d_{model}$ | $d_{ff}$ | $n_{heads}$ | $n_{layers}$ | $B$ | $T$ |
|---|---|---|---|---|---|---|
| 1.07 | 2048 | 8192 | 16 | 16 | 352 | 27743 |
| 0.54 | 1280 | 9048 | 10 | 13 | 352 | 39464 |
| 0.43 | 1280 | 9472 | 10 | 10 | 512 | 21696 |
| 0.21 | 960 | 9368 | 15 | 7 | 128 | 76293 |

**Table 5**  MoE model configurations. Model size is measured in billions of parameters (B). In columns, $n_{exp}$ = number of experts, $d_{exp}$ = per-expert hidden size, SED = selected expert dimension, $B$ = batch size, $T$ = total iterations.

| Model Size (B) | $n_{exp}$ | $d_{exp}$ | SED | $B$ | $T$ |
|---|---|---|---|---|---|
| 2.16 | 8 | 2888 | 8664 | 32 | 30517 |
| 2.15 (Green) | 88 | 352 | 704 | 32 | 30517 |
| 2.15 (Orange) | 89 | 352 | 352 | 32 | 30517 |

### C.2  FIGURE 7 AND TABLE 2

We give the comprehensive table of all datapoints from (Besiroglu et al., 2024), where the linear regression fits well in all cases.

| model size(B) | num of horizons | $2\tilde{q}_1\tilde{q}_2$ | $\tilde{L}_\infty$ | $R^2$ score |
|---|---|---|---|---|
| 0.074 | 5 | 3.22e+04 | 2.825 | 0.991 |
| 0.090 | 3 | 3.19e+04 | 2.774 | 0.991 |
| 0.106 | 4 | 3.38e+04 | 2.706 | 1.000 |
| 0.117 | 3 | 3.27e+04 | 2.692 | 0.996 |
| 0.140 | 7 | 3.04e+04 | 2.670 | 0.991 |
| 0.163 | 3 | 3.11e+04 | 2.619 | 1.000 |
| 0.175 | 7 | 3.08e+04 | 2.619 | 0.995 |
| 0.196 | 4 | 3.14e+04 | 2.582 | 0.999 |
| 0.217 | 6 | 3.54e+04 | 2.526 | 0.998 |
| 0.251 | 3 | 3.37e+04 | 2.517 | 1.000 |
| 0.279 | 8 | 3.29e+04 | 2.498 | 0.999 |
| 0.305 | 7 | 3.14e+04 | 2.488 | 0.997 |
| 0.425 | 8 | 3.27e+04 | 2.430 | 0.998 |
| 0.489 | 4 | 3.30e+04 | 2.404 | 0.999 |
| 0.552 | 8 | 3.24e+04 | 2.382 | 0.999 |
| 0.586 | 8 | 3.25e+04 | 2.368 | 0.994 |
| 0.632 | 8 | 3.17e+04 | 2.367 | 0.998 |
| 0.664 | 3 | 3.46e+04 | 2.330 | 0.999 |
| 0.724 | 3 | 3.53e+04 | 2.320 | 0.999 |
| 0.817 | 10 | 3.28e+04 | 2.315 | 0.994 |
| 0.893 | 3 | 3.35e+04 | 2.304 | 0.998 |
| 1.019 | 7 | 3.06e+04 | 2.305 | 0.997 |
| 1.143 | 10 | 3.10e+04 | 2.275 | 0.998 |
| 1.265 | 10 | 3.05e+04 | 2.286 | 0.986 |
| 1.426 | 3 | 4.07e+04 | 2.214 | 0.984 |
| 1.429 | 9 | 3.18e+04 | 2.253 | 0.996 |
| 1.592 | 4 | 4.22e+04 | 2.182 | 0.997 |
| 1.611 | 9 | 3.36e+04 | 2.228 | 0.995 |
| 1.730 | 7 | 3.53e+04 | 2.207 | 0.998 |
| 1.795 | 11 | 3.41e+04 | 2.211 | 0.997 |
| 2.004 | 8 | 3.62e+04 | 2.178 | 0.999 |
| 2.280 | 7 | 4.41e+04 | 2.128 | 1.000 |
| 2.636 | 6 | 4.08e+04 | 2.113 | 0.998 |
| 2.979 | 10 | 5.90e+04 | 2.016 | 0.990 |
| 4.519 | 6 | 3.83e+04 | 2.106 | 0.978 |
| 6.792 | 8 | 4.66e+04 | 2.023 | 0.999 |
| 9.290 | 4 | 4.29e+04 | 2.046 | 0.988 |
| 12.560 | 3 | 4.23e+04 | 2.053 | 1.000 |

**Table 6**  Comprehensive version of Table 2.

# D   ADDITIONAL EXPERIMENTS

## D.1   EFFECT OF OVERFITTING

Our experiments are primarily conducted when the train loss and test loss are similar, without much overfitting. For example, our GPT2 runs in Section 5 use up to 260B tokens (about 30 epochs of OpenWebText) and test loss is close to train loss.

In order to validate our generalization in the overfitting regime, we train ViT models and ResNet models on ImageNet for multiple runs up to 1 epoch, and a ResNet50 on CIFAR10 for multiple runs up to 820 epochs. As expected, within 1 epoch when overfitting is insignificant, $O(1/\sqrt{T})$ convergence holds for both train loss and test loss, although the coefficients of fitting may be slightly different. However, with overfitting (e.g. after 50 epochs), we observe that $O(1/\sqrt{T})$ convergence only holds for train loss but no longer for test loss (see the last plot in Figure 12).

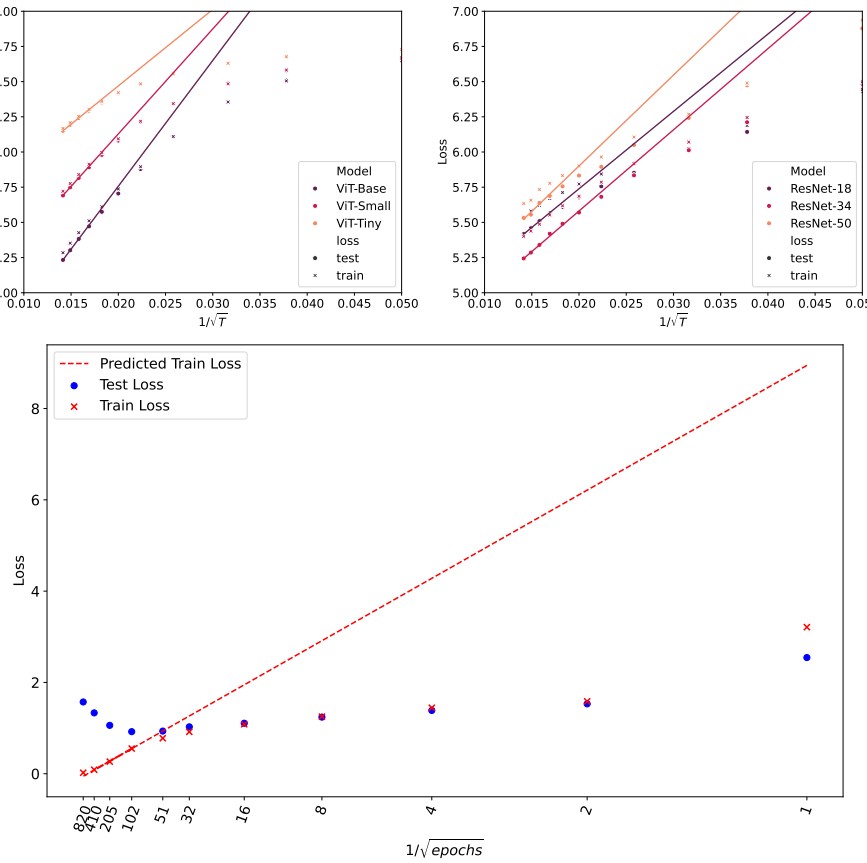

**Figure 12**  Loss of multiple runs with scaled peak learning rate and AdamW. Upper left: ViT model on ImageNet (within 1 epoch). Upper right: ResNet models on ImageNet (within 1 epoch). Lower: ResNet50 on CIFAR10 (over 1 epoch), where we have marked the number of epochs along x-axis.

All models are optimized by AdamW with $\beta_1 = 0.9, \beta_2 = 0.999$, weight decay$= 0.01$, batch size 128. Learning rate schedule is cosine decay with peak learning rate $0.01/\sqrt{T}$.

## D.2 Parameter-efficient v.s. full model training

We experiment with full model training and parameter-efficient fine-tuning of GPT2 models. We test small/medium/large sizes on E2E dataset. For parameter-efficient fine-tuning, we apply low-rank adaptation (LoRA) with rank 4. Our experiments in Figure 13 show that $O(1/\sqrt{T})$ convergence holds within both training regimes. All models are optimized by AdamW with $\beta_1 = 0.9, \beta_2 = 0.999$, weight decay= 0.01, batch size 8. Learning rate schedule is linear with peak learning rate $0.0005/\sqrt{T}$ for LoRA and $0.00005/\sqrt{T}$ for full model fine-tuning.

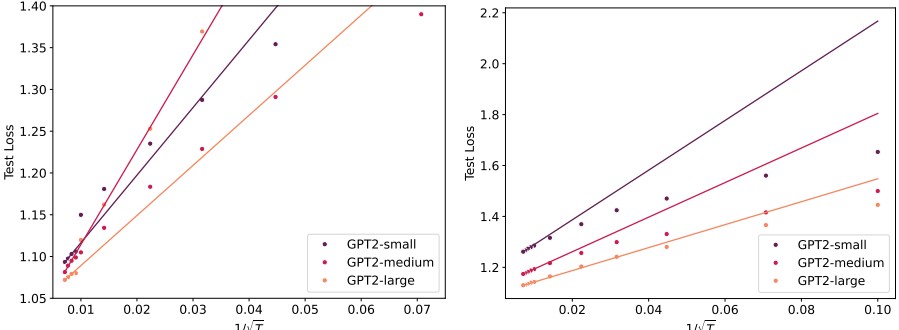

**Figure 13** Loss of multiple fine-tuning runs with scaled peak learning rate and AdamW, on E2E dataset. Left: full model fine-tuning. Right: LoRA fine-tuning.

## D.3 Ablation study on hyperparameters

We further evaluate our Generalization 4 through the ablation study over five key hyperparameters, such as weight decay, gradient clipping, momentum coefficient, batch size, and random seeds. For each new row in Table 7, we launch 5 runs with $T \in [60^2, 70^2, 80^2, 90^2, 100^2]$, so in total we summarize the statistics of 35 runs. We fit linear regression of validation loss against $1/\sqrt{T}$, and we consistently observe goodness of fit.

|  | $\tilde{Q}$ estimation (std err) | $\tilde{L}_\infty$ estimation (std err) | $R^2$ fit score |
|---|---|---|---|
| default (seed=1337) | 23.9094 (0.469) | 2.8298 (0.004) | 0.998 |
| seed=3333 | 24.1820 (0.220) | 2.8321 (0.003) | 1.000 |
| seed=8888 | 25.0086 (0.914) | 2.8146 (0.012) | 0.995 |
| default (batch size=512) | 23.9094 (0.469) | 2.8298 (0.004) | 0.998 |
| batch size=64 | 39.5403 (1.238) | 2.9670 (0.016) | 0.996 |
| default (clipping=0.0) | 23.9094 (0.469) | 2.8298 (0.004) | 0.998 |
| clipping=1.0 | 24.3836 (0.481) | 2.8207 (0.006) | 0.998 |
| default (momentum=0.95) | 23.9094 (0.469) | 2.8298 (0.004) | 0.998 |
| momentum=0.9 | 23.8508 (0.497) | 2.8432 (0.007) | 0.998 |
| default (weight decay=0.01) | 23.9094 (0.469) | 2.8298 (0.004) | 0.998 |
| weight decay=0.0 | 25.1154 (0.416) | 2.8158 (0.005) | 0.999 |

**Table 7** Ablation study of Generalization 4 over weight decay, gradient clipping, momentum coefficient, batch size, and random seeds. High $R^2$ score supports our $O(1/\sqrt{T})$ loss convergence with $1/\sqrt{T}$-scaled learning rate.

Here, the default setting is $\eta_{\text{ref}} = 1.0$ for GPT2 0.1B, batch size 512(*1024 context length), weight decay 0.01, momentum 0.95, random seed 1337, clipping norm 0.0, Muon-NSGD optimizer and cosine learning rate schedule (with 2% warmup), the same as in Section 5.

