# OpenReview forum: "Convex Dominance in Deep Learning I: A Scaling Law of Loss and Learning Rate"
_ICLR.cc/2026/Conference — ICLR 2026 Poster_

### Official Review · Reviewer_CJJk · 2025-10-27

**Soundness:** 3
**Presentation:** 2
**Contribution:** 2
**Rating:** 6
**Confidence:** 4

**Summary:**

This paper derives the dependency of the last-iterate loss on the sequence of the learning rate for the SGD algorithm under the assumption of convexity and bounded gradient. From the fine-grained upper bound based on the specific learning rate values, the paper derives $i)$ asymptotic bounds on the loss that depends on the peak learning rate, the optimal convergence rate, as well as a qualifying exam of the learning rate schedule that achieves the optimal convergence rate. The paper also presented an extension of the regulating dynamic to the setting of deep learning, and observed empirically the coincidence between the theoretical prediction and the experimental convergence. Lastly, the paper presented a scaling law governing the model size, the number of iterations, and the reference learning rate.

**Strengths:**

1. The result of the paper applies to any learning rate schedules, rather than just the cosine and WSD in Schaipp et al., 2025.
2. The paper empirically verified the similarity of the empirical training behavior of deep learning models and the theoretically derived last-iterate convergence.
3. The paper also derived a qualifying exam for the learning rate schedule and a scaling law of the learning rate schedules.

**Weaknesses:**

1. The paper assumes a uniform bound on the expected gradient, which in many case does not hold (e.g. in the training of neural networks based on MSE loss). The following theoretical bound are based on this assumption, which is probably the reason why in the experiment part the paper has to adopt the Adam optimizer instead of the regular SGD (to condition out the influence of the gradient norm). In the setting of the deep learning extension, this uniform upper bound factor $G$ is modified to be a more sophisticated term that may not solely depend on the gradient norm, which can be even harder to track.
2. The qualifying exam of the learning rate schedules takes a complicated form that requires several integrations. This can be hard to apply to a general learning rate schedules. The paper can consider to incorporate several examples that explicitly apply this qualifying exam and validate the result using experiments.
3. The dependency on $\eta_{\text{ref}}$ is not discussed thoroughly in the paper. Although is can be understood that the paper's focus is on the scaling of $T$, it is certainly the case that not all choices of $\eta_{\text{ref}}$ will lead to a meaningful convergence, which is not accounted for in this paper.
4. Some technical part lacks clarity. For instance, it is not clear how to obtain Eq. (2.4) from Eq. (2.3). Moreover, the definition of $N_{\text{small}}$ and $T_{\text{small}}$ in the scaling law part is not clear.

**Questions:**

None

---

> ### Author Response · Authors · 2025-11-18
>
> Thank you for your discussion! We would like to respond to your feedback point-to-point. If you are satisfied with our response, we would be grateful if you could consider raising your score.
>
> *---The paper assumes a uniform bound on the expected gradient, which in many case does not hold (e.g. in the training of neural networks based on MSE loss). The following theoretical bound are based on this assumption, which is probably the reason why in the experiment part the paper has to adopt the Adam optimizer instead of the regular SGD (to condition out the influence of the gradient norm).*
>
> We agree that the uniform bound may not hold in many cases, including training neural networks on cross-entropy or MSE loss. Nevertheless, we may only need a bound on the optimization trajectory ($\exists G$ s.t. $E|g(w_t)|^2 \leq G^2$ for $w_1,w_2,...w_T$), instead of a uniform bound ($\exists G$ s.t. $E|g(w)|^2 \leq G^2$ for all $w$). We admit even this weaker assumption is hard to track but it may be practically useful, e.g. if we project the neural network's parameters into a domain, like projected gradient descent.
>
> Experiment-wise, we kindly remind the reviewer that we have experimented with SGD in Figure2 (ResNet model and ImageNet dataset).
>
> *---The paper can consider to incorporate several examples that explicitly apply this qualifying exam and validate the result using experiments.*
>
> We agree with the reviewer and this is exactly our Theorem 1 (discussed right after Condition 2.4 and proved in Appendix A). We have applied this qualifying exam to 5 lr schedules and showed that "linear decaying, cosine decaying, and WSD schedules
> indeed pass the qualifying exam, whereas the constant and square-root inverse schedules fail." Please let us know if this addresses your concern.
>
> *---Some technical part lacks clarity. For instance, it is not clear how to obtain Eq. (2.4) from Eq. (2.3). Moreover, the definition of $N_{small}$ and $T_{small}$ in the scaling law part is not clear.*
>
> We agree it is hard to directly go from (2.3) to (2.4) without reading the reference, i.e. Defazio et al., 2023, Corollary 12. However, we hope the reviewer would understand that it took a long analysis in that paper to derive the bound, and hence it is beyond the scope of this work to reproduce that. In the updated version, we have explained $N_{small}$ as any smaller model with $N_{small}\leq N$, and $T_{small}$ as any shorter training horizon with $T_{small}\leq T$, so that we can use small-scale training to inform the large-scale training's hyperparameter.

---

> > ### Comment · Reviewer_CJJk · 2025-11-26
> >
> > Thank you for your rebuttal. I found the answers convincing to me. Could you also elaborate on my comment about $\eta_{\text{ref}}$ above?

---

> > > ### Author Response · Authors · 2025-11-27
> > >
> > > Sure! Empirically speaking, we need to determine $\eta_{ref}$ by a sweep, e.g. in Figure 8 we sweep over {0.01,0.03,0.1,0.3,1.0,3.0} and in Figure 9 we sweep over {0.01,0.03,0.1,0.3,1.0,3.0,10.0,30.0}. Too large $\eta_{ref}$ may not converge unless the training horizon T is very large, so that the peak learning rate $\eta_{ref}/\sqrt{T}$ may still be usable. Additionally, the dependency on $\eta_{ref}$ can be transferred across model sizes in Figure 10, e.g. from 0.1B to 7B models.

---

### Official Review · Reviewer_cgXX · 2025-10-31

**Soundness:** 3
**Presentation:** 2
**Contribution:** 3
**Rating:** 4
**Confidence:** 3

**Summary:**

The paper proposes a series of upper bounds that are used to show—through regression fitting of their parameters—that different combinations of deep learning models and optimization algorithms display a convex-like behavior across the training path. In particular, this is used to predict asymptotic $O(\frac{1}{\sqrt{T}})$ loss convergence and learning rate. The upper bounds are characterized as “generalizations” of bounds derived from loss functions that otherwise are convex (or at least “star-convex”) and have bounded gradient.

**Strengths:**

The paper strengths are:
- The paper’s motivation of predicting convex-like behavior is relevant in the understanding of the training of deep learning models.
- The scaling laws found by the paper, which establishes evidence towards convex-like behavior in the training of deep models, are extensively characterized in their data-driven approach.
- The paper has enough models and training procedures to properly demonstrate its claims. I really appreciate that from the authors.

**Weaknesses:**

Although the paper’s topic is relevant, there were multiple things that were unclear in the paper’s presentation (some of them could have been spotted after diligent proofreading). Also, there were confusing parts in the presentation of both theoretical and experimental results—I will detail this below. Given that there is still work to be done to improve the paper, I am giving the current score.

**>>Important things:**
- I find it odd that in the last sentence of the paragraph from line 057, there is reference to equations (2.3) and (2.4) and the work by Defazio et al. (2023), as if the reader could immediately recognize what the authors are referring to with such references. Moreover, it is hard to know from the sentence in lines 059-061 what exactly the authors are doing differently than Defazio et al. (2023).
- The inclusion of “Example 1.1.” is confusing. A few notes:
  - There is no explanation of the meaning of parameters $L_*$, $D$, $T$, and $G$. How is the reader supposed to understand Example 1.1 then?
  - Moreover, there is another problem: it is said that the shown equation “aligns with the empirical trade-off in deep learning that larger $\eta$ converges faster but to a higher loss, and vice versa”. Two problems with this: Firstly, **there is no citation** to back up this claim: the authors must add one. Secondly, the statement is misleading: what becomes higher is the **upper bound on the loss** that the authors have derived, and not the loss *itself* (but the reader has no way to know this, since no explanation is given in Example 1.1.).
  - Example 1.1 needs to be urgently fixed considering what I pointed out. It may need to be fully reformulated. Also, the title “Example” sounds odd–maybe “Finding” or even "Example of Findings” could be better.
- Line 116: Shouldn’t $\eta_{peak}$ be constrained to be positive, instead of being any real number?
- Equation (2.2): It should say “$\exists G$ such that $\forall w”.
- Remark 2.2: $w_*$ is stated as being **unique** because of the expression $w_*=argmin$ (instead of using the expression $w_*\in argmin$). However, convex functions **do not necessarily** have a unique minimum (I don’t think that Condition 2.1 implies uniqueness; please, correct me if I am wrong). Why is uniqueness assumed? Does uniqueness play an important role in **any** of the derivations in the paper?
- The paper goes through a series of elemental derivations to obtain equation (2.3). However, then the work (Defazio et al., 2023) is cited to obtain (2.4), without any mention of whether equation (2.3) was used to derive equation (2.4), and, if that is the case, there is no mention of how (2.4) came from (2.3). The question is: why derive equation (2.3) when at the end, as the paragraph that follows equation (2.4) states, only (2.4) is going to be used in the whole paper? Why not simply cite (Defazio et al., 2023) and avoid derivations that do not seem useful at all? The shown derivations would only be useful if an explicit connection and derivation is made between (2.3) and (2.4), but this is absent in the paper.
- In line 160 and in Table 1, the term “optimal loss” **does not** refer to the optimal loss itself, but it refers to the “**optimal upper bound on the loss**”. This needs to be fixed.
- Line 185: “Section 2.4” is mentioned when, in fact, the paper has no “Section 2.4”. What is supposed to go there?
- It is confusing that equation (2.2) of Condition 2.1 is included as an assumption for the **whole** paper, when in reality, according to Remark 2.2., only star-convexity is needed (the unnumbered equation inside Remark 2.2). So, if all the paper results only need star-convexity (assuming Defazio et al (2023) also used star convexity), why not replace equation (2.2) by star-convexity? One could just make a simple remark saying that convexity is a stronger notion than star-convexity. Do the authors have a strong reason for keeping (2.2) inside Condition 2.1 instead of being replaced by star convexity?
- I **strongly suggest to emphasize** in Section 3 that: **>>>>** All the results described in Section 3.2 and Section 3.3 indicate convex-like behavior, and that such behavior is due to **two factors**: the model itself and the chosen optimization algorithm. These two factors were originally present in the upper bound where the generalized bound was derived from. In other words, though the loss function of the neural model is non-convex, its behavior **along the optimization path** behaves according to a convex-like upper bound. **<<<<**. This summarization is currently absent in the important Section 3. Also, the use of the term “optimization path” as *where* the convex-like behavior happens is very useful.
- There seem to be figures missing or a mismatch between the text and the figures caption. For example, line 261 mentions that ResNet50 is present in Figure 2, however, Figure 2’s caption shows ResNet18. Likewise, line 282 mentions that ReNet18 is present in Figure 3, while Figure 3’s caption shows ResNet50. In any case, the paper **will greatly benefit** from the presentation of both SGD and AdamW for both ResNet18 and ResNet50 models.
- Section 4.1 does not mention which model is being trained for the experiments. I suspect that it is some sort of language model, but the model **must** be included.
- It is really cumbersome for the reader to have all types of generalizations spread out in the paper, when they are sort of related to each other. I **strongly suggest** to put all generalization bounds on a single table to aid the reader to better understand the paper. The table can include columns **describing** (i) where/when each generalization bound is used, and (ii) which parameters need to be obtained through regression.
- To better understand the experiments of Figure 8 and Figure 9 (something similar for Figure 10), it is important to include a text explaining that the reason why one cares about **fitting straight lines** is because the upper bound in Generalization 4 is just a line: for different values of $\frac{1}{\sqrt{T}}$, the slope $\tilde{Q}$ and the offset $\tilde{L}_*$ needs to be estimated. Unless I missed something, this explicit information of which **parameters the figures are fitting** is missed in Section 5.
- Many citations are **missing their publication year**. This needs to be fixed.

**>>Unclear things:**
- Lines 031-032: How is it that the Llama training being mentioned is similar to the other SGD training being mentioned? What kind of metric of similarity was used? Which citation presents such a claim?
- Line 076: uses the term “qualifying exam” without any reference of what this means and with respect to *what* something is being qualified. Moreover, the term “qualified” is used again in line 105 without any explanation of its meaning. The reader has to read past two pages to understand what these expressions mean. These expressions need to be specified early on if they are going to be used.
- Line 183: refers to Theorem 1. The problem is that Theorem 1 is in the Appendix, which makes the paper less self-contained. It would help to include a little description of the specific cases that Theorem 1 analyzes: in other words, to move the explanations from lines 216-217 to inside the sentence that starts in line 183.
- The important information that only half of the iterations are used to fit the bound in page 5 should be moved from footnote 1 to the main text of Section 3.2. This is an important experimental detail which is odd to relegate to a footnote.

**>>Other presentation issues:**
- Many references need to have parenthesis before the names of the cited authors; this is used when one refers to the *work* and not to the *authors themselves*. Use “\citep{}” for this. This occurs throughout the paper and needs attention. The first time this problem appears, for example, is in lines 026-027 (it should start with “(Garipov et al, 2028; …)”). Similarly, it should be “(Krogh & …, 1991)” and “(Hoerl & …, 1970)” in lines 054-055. Multiple other parts to fix exist throughout the paper.
 - Lines 045-046: How is it that the spectral properties of the Hessian indicate “convexity”? Does it have to do with local curvature or something related to it? Please explain.
- End of line 088: should say “for the fixed value $\alpha=0.5$”.
- Section 2.1: specify that $g_t:=g(w_t)$.
- Line 140: it should say “applying Jensen’s inequality”.
- Line 158-159: it should say “We show the upper bounds on the loss in terms of different learning rates (...)”.
- Line 160: it says “see appendix”. Which section of the appendix does it refer to?
- Line 161: it should say “in Table 1” instead of using the preposition “from”.
- Line 430: include citation from where Muon comes from.

**Questions:**

Please, see the Weaknesses section.

---

> ### Author Response · Authors · 2025-11-18
>
> Thank you for writing such a careful and diligent review! We will try our best to address your questions. We strive to cover all your points, although typos and notation issues are directly modified without a written response below. Please let us know if our revision is satisfactory.
>
> *---I find it odd that in the last sentence of the paragraph from line 057, there is reference to equations (2.3) and (2.4) and the work by Defazio et al. (2023), as if the reader could immediately recognize what the authors are referring to with such references. Moreover, it is hard to know from the sentence in lines 059-061 what exactly the authors are doing differently than Defazio et al. (2023).*
>
> We modify our line 59-61 as "Our work is directly based on Corollary 12 of Defazio et al. (re-stated in (2.4)), and a simplified version can be found in (2.3), which already provides some insights as summarized in the following example."
>
> We provide the simple bound (2.3) to be self-contained, but (2.4) characterizes the loss more precisely, despite that it is much harder to derive. Even though we didn't reproduce the analysis that leads to (2.4), we want to give enough credit to Defazio et al. (2023) by citing their result early in the introduction. To be clear, we didn't modify (2.4) but we leverage it to provide concrete forms for specific learning rate schedules and a practical qualifying exam, which eventually leads to 1/sqrt(T) scaled learning rate and O(1/sqrt(T)) loss convergence.
>
> *---There is no explanation of the meaning of parameters. How is the reader supposed to understand Example 1.1 then? ...Also, the title “Example” sounds odd–maybe “Finding” or even "Example of Findings” could be better.*
>
> We agree that some notations are defined later in Section 2, hence we move this example after Equation 2.3 and change the title to "Finding".
>
> *---it is said that the shown equation “aligns with the empirical trade-off in deep learning that larger $\eta$ converges faster but to a higher loss, and vice versa”. Two problems with this: Firstly, there is no citation to back up this claim: the authors must add one. Secondly, the statement is misleading: what becomes higher is the upper bound on the loss that the authors have derived, and not the loss itself (but the reader has no way to know this, since no explanation is given in Example 1.1.).*
>
> We have added a citation to back up this claim and we agree there is an assumption that the upper bound is close to the actual loss, which is made implicit for the sake of brevity. We hope this assumption is empirically justified in later experiments, e.g. Figure 2 to 5.
>
> *---Remark 2.2: $w_*$ is stated as being unique because of the expression  (instead of using the expression ). However, convex functions do not necessarily have a unique minimum (I don’t think that Condition 2.1 implies uniqueness; please, correct me if I am wrong). Why is uniqueness assumed? Does uniqueness play an important role in any of the derivations in the paper?*
>
> We agree we don't need uniqueness. This expression has been corrected.
>
> *---why derive equation (2.3) when at the end, as the paragraph that follows equation (2.4) states, only (2.4) is going to be used in the whole paper? Why not simply cite (Defazio et al., 2023) and avoid derivations that do not seem useful at all? The shown derivations would only be useful if an explicit connection and derivation is made between (2.3) and (2.4), but this is absent in the paper.*
>
> We included (2.3) to be more self-contained and readable, so the authors can understand at least the first three terms in (2.4). However, we hope the reviewer would understand that it took a long analysis in
> that paper to derive the bound, and it is too lengthy to include in this work. Please let us know if we need to move (2.3) to the appendix.
>
> *---It is confusing that equation (2.2) of Condition 2.1 is included as an assumption for the whole paper, when in reality, according to Remark 2.2., only star-convexity is needed (the unnumbered equation inside Remark 2.2). So, if all the paper results only need star-convexity (assuming Defazio et al (2023) also used star convexity), why not replace equation (2.2) by star-convexity? ... Do the authors have a strong reason for keeping (2.2) inside Condition 2.1 instead of being replaced by star convexity?*
>
> Firstly, can we confirm if the reviewer is actually referring to (2.1), not (2.2)? We currently keep convexity for readability, because convexity is a more commonly known concept than star-convexity. Another reason to keep (2.1) is to explain (2.3): convexity is used twice to derive the simple loss bound. The first time is to get $L_t-L_*\leq \eta (w_t-w_*)g_t$, which can be alternatively obtained through star-convexity; the second time is to apply Jensen's inequality, so that we get the bound on averaged iterate in (2.3). However, we are actually interested in (2.4) the last iterate, hence we don't need Jensen's inequality and convexity.

---

> ### Author Response · Authors · 2025-11-18
>
> *---Line 185: “Section 2.4” is mentioned when, in fact, the paper has no “Section 2.4”. What is supposed to go there?*
>
> This is to our shock, and there may be a system error! We do have Section 2.4 from line 201 to line 228 in our original submission (and the current one). Could the reviewer double-check and let us know if we need to escalate to OpenReview?
>
>
>
> *---I strongly suggest to emphasize in Section 3 that: >>>> All the results described in Section 3.2 and Section 3.3 indicate convex-like behavior, and that such behavior is due to two factors: the model itself and the chosen optimization algorithm. These two factors were originally present in the upper bound where the generalized bound was derived from. In other words, though the loss function of the neural model is non-convex, its behavior along the optimization path behaves according to a convex-like upper bound. <<<<. This summarization is currently absent in the important Section 3. Also, the use of the term “optimization path” as where the convex-like behavior happens is very useful.*
>
> We agree on the optimization path, but we don't see how the model (architecture?) indicates the convex-like behavior. The empirical applicability holds for multiple neural models and the convex theory does not take the architecture into account. We would love to extend this discussion and add it to the end of Section 3.1 in the camera-ready version.
>
> *---There seem to be figures missing or a mismatch between the text and the figures caption. For example, line 261 mentions that ResNet50 is present in Figure 2, however, Figure 2’s caption shows ResNet18. Likewise, line 282 mentions that ReNet18 is present in Figure 3, while Figure 3’s caption shows ResNet50. In any case, the paper will greatly benefit from the presentation of both SGD and AdamW for both ResNet18 and ResNet50 models.*
>
> Thank you for pointing out this typo. We indeed experimented with ResNet18. This has been corrected in the main text.
>
> *---Section 4.1 does not mention which model is being trained for the experiments. I suspect that it is some sort of language model, but the model must be included.*
>
> We have polished Section 4.1 to give more information on the models. They are language models without specific names. We have included some architecture details in Appendix C.1.
>
> *---I strongly suggest to put all generalization bounds on a single table to aid the reader to better understand the paper. The table can include columns describing (i) where/when each generalization bound is used, and (ii) which parameters need to be obtained through regression.*
>
> We have added the requested table in Appendix A.2, which is introduced in the first paragraph of the "Contributions" section.
>
> *---To better understand the experiments of Figure 8 and Figure 9 (something similar for Figure 10), it is important to include a text explaining that the reason why one cares about fitting straight lines is because the upper bound in Generalization 4 is just a line: for different values of... the slope and the offset needs to be estimated. Unless I missed something, this explicit information of which parameters the figures are fitting is missed in Section 5.*
>
> We have added text to explain that $\tilde L_\infty$ and $\tilde Q$ are the parameters to be fitted and that Generalization 4 can be visualized as a straight line that we indeed observe throughout Section 5.
>
>
> *---Many citations are missing their publication year. This needs to be fixed.*
>
> All missing years are added.
>
> *---Lines 031-032: How is it that the Llama training being mentioned is similar to the other SGD training being mentioned? What kind of metric of similarity was used? Which citation presents such a claim?*
>
> We have rewritten line 31 as "Llama training (non-convex) with AdamW is closely similar to convex
> optimization with SGD, in terms of the shapes of loss curves in Figure 1 of \_cite[Schaipp et al].
> "
> in which the citation is "The Surprising Agreement Between Convex Optimization Theory and Learning-Rate Scheduling for Large Model Training".
> *---The important information that only half of the iterations are used to fit the bound in page 5 should be moved from footnote 1 to the main text of Section 3.2. This is an important experimental detail which is odd to relegate to a footnote.*
>
> We agree with the reviewer and have moved it to the first paragraph of Section 3.2.
>
> *---How is it that the spectral properties of the Hessian indicate “convexity”? Does it have to do with local curvature or something related to it?*
>
> We have polished the statement that claims if all eigenvalues of the Hessian at a point are positive, then the loss is locally convex around this point. Please see line 42.

---

### Official Review · Reviewer_yGzE · 2025-10-31

**Soundness:** 3
**Presentation:** 3
**Contribution:** 3
**Rating:** 8
**Confidence:** 3

**Summary:**

This paper extends the discoveries in [1], which states that the upper bound driven from convex optimization for SGD matches the loss curve of deep learning in general. In other words, in deep learning SGD has coeffients A, B that satisfy
$$
L(w_T) - L^{*} \leq \frac{A}{T\eta_{max}} + B\eta_{max},
$$
where $\eta$ is learning rate, $A$ is an analogy of the diameter, and $B$ is an analogy of the maximal function value. The core contribution of this paper is extending this tendancy to other optimizers and find that the principle holds indeed. Different from SGD, the coefficients are not exactly given from convex optimization, so they propose a "data-driven" approach where they use certain points on the training curve to approximate $A$, $B$, and predict the rest. They validate their discoveries across various experiments: the tendancy across different max learning rates, upon the optimal learning rate, and a scaling law that predicts both loss and optimal learning rate.

[1] Schaipp, Fabian, et al. "The surprising agreement between convex optimization theory and learning-rate scheduling for large model training." arXiv preprint arXiv:2501.18965 (2025).

**Strengths:**

I believe it is rare to have a nice, predictable tendancy in neural network training, and such discovery as in this paper is useful because once we can predict what will work best in practice, e.g. can compute the optimal learning rate by hand. Even though this is an extension (at least in my view), the fact that the tendancy exists for different optimizers is an important discovery, because it is not obvious. So one strength of this paper is that it proposes a property of neural network training that seems actually useful.

Another strength of this paper is an extensive experimental result with very high correlation ($R^2 \geq 0.95$, mostly even higher). I think these results show a strong signal that the authors have indeed identified a tendancy.

**Weaknesses:**

One thing that was not clear to me was:

so we have an upper bound that looks like
$$
L(w_T) - L^{*} \leq \frac{A}{T\eta_{max}} + B\eta_{max}.
$$

So is $A, B$ a function of the optimizer, the model, and the learning rate schedule? In other words, if these three are fixed, is $A$ and $B$ fixed regardless of the maximal learning rate? If this is not the case, I think it would make the paper much weaker (and there is potential that I may lower the score), because even though there is a tendancy there is no practical benefit as for every maximal learning rate we should find $A$, $B$ again and there might not be a practical scheme. It would be great if the authors clarify this point, and it would be amazing to show a practical application of this result in the experiments section (e.g. actually tuning learning rate with the prediction).

Eventually, it would be good to explicitly show how $A, B$ and the various different factors in training is related.

**Questions:**

One thing I got curious: why is this phenomenon happening? Is it really because the landscape becomes convex in the latter parts of training? If that is the case, is there a way to actually verify whether the landscape is convex or not?

---

> ### Author Response · Authors · 2025-11-18
>
> Thank you for your review and the interesting questions!
>
> *---So is A,B a function of the optimizer, the model, and the learning rate schedule? In other words, if these three are fixed, is A and B fixed regardless of the maximal learning rate?*
>
> % We would like to clarify that the predictable loss ("practical benefit") doesn't have to come from your condition, i.e., A, B should be fixed for any peak eta. Actually, we
>
> We would like to clarify that we can predict the loss regardless of the maximal learning rate (without re-tuning), **if the maximal learning rates are scaled by $1/\sqrt{T}$**. Looking at
> $$L(w_T)-L_\infty\leq \frac{A}{T\eta_{peak}}+B\eta_{peak}$$
> we need to ensure multiple maximal learning rates are from the same family of $\eta_{peak}=\eta_{ref}/\sqrt{T}$. For example in Figure 9, three of the runs are $(\eta_{peak}=10/\sqrt{T}, T=2500), (\eta_{peak}=10/\sqrt{T}, T=6400), (\eta_{peak}=10/\sqrt{T}, T=10000)$. These runs have different maximal learning rates but are from the same $\eta_{ref}$.
>
> Replacing $\eta_{peak}={\eta_{ref}}/{\sqrt{T}}$ gives our Generalization 4:
> $$L(w_T)-L_\infty\leq \frac{A}{T\eta_{peak}}+B\eta_{peak} = \frac{A/\eta_{ref}+B\eta_{ref}}{\sqrt{T}}:=Q(\eta_{ref})/\sqrt{T}$$
> That is, the numerator (as a function of A and B; we denote it as $Q(\eta_{ref})$) is fixed even though the maximal learning rates are different. This can be confirmed by the fit of straight lines in Figure 8 and Figure 9 (e.g. blue dots in the right-most sub-plots). Notice that we do not need to estimate A and B separately.
>
> This offers the practical benefit that we don't need to find A,B (or Q) for every maximal learning rate. All we need are short runs ($T<3000$ on small models) to tune $\eta_{ref}^*$ and then automatically determine the maximal learning rate for any large $T$ (up to 80 times in scale) and on large models (up to 70 times in scale).
>
> On the other hand, by comparing blue and orange lines in Figure 10, we see Q depends on optimizers as the slopes are different for AdamW and Muon-NSGD; by comparing orange, green and red lines in Figure 10, we see Q depends on model sizes as the slopes are also different.
>
> *---why is this phenomenon happening? Is it really because the landscape becomes convex in the latter parts of training? If that is the case, is there a way to actually verify whether the landscape is convex or not?*
>
> To be honest, it remains a mystery why this phenomenon happens and we admits this limitation in our "Conclusion" section. As we discussed in Remark 2.2, what we are observing is not necessarily convexity, e.g. it can be star-convexity. Currently, researchers have some evidence about convexity in the small model (fewer than 1M param by looking at full Hessian matrix, e.g. "An Investigation into Neural Net Optimization via Hessian Eigenvalue Density" and "Eigenvalues of the Hessian in Deep Learning: Singularity and Beyond") or in low-dimension (e.g. along the gradient dimension like "Gradient descent with generalized Newton's method"), but it is challenging to verify the landscape of large model in high-dimension.

---

> > ### Comment · Reviewer_yGzE · 2025-11-26
> >
> > Dear authors,
> >
> > Thank you for your comment. I think my initial assessment was fair and I maintain my score.

---

### Official Review · Reviewer_WNR5 · 2025-11-01

**Soundness:** 3
**Presentation:** 3
**Contribution:** 2
**Rating:** 6
**Confidence:** 4

**Summary:**

This paper explores whether convex optimization insights can reliably predict and control loss dynamics in deep learning through learning-rate (LR) schedules. Starting from a known convex, bounded-gradient last-iterate bound, the authors compute schedule-specific last-iterate bounds and show that horizon-aware “qualified” schedules (e.g., linear, cosine, WSD) achieve $\mathcal{O}\left(\tfrac{1}{\sqrt{T}}\right)$ optimal last-iterate convergence with $\eta_{\text{peak}} \propto \tfrac{1}{\sqrt{T}}$ (Theorem 1, Table 1). They then propose a training-free “qualifying exam” (Condition 2.4) using integral approximations to identify such schedules. Moving beyond convex SGD, they introduce data-fitted coefficients $(\tilde{q}\_1, \tilde{q}\_2, L\_{\infty})$ and show empirically that (i) a sequence-to-sequence mapping from LR schedule to loss captures training trajectories across models, optimizers, and schedules ($R^2 \geq 0.95$ in most cases), and (ii) last-iterate loss scales approximately linearly in $1/\sqrt{T}$ under $\eta_{\text{peak}} \propto 1/\sqrt{T}$, with cross-horizon and cross-size extrapolations. Analyses include new runs in vision and language and external large-scale results to support a simple two-dimensional scaling relation.

**Strengths:**

- **Broad, consistent empirical signal.** The sequence-to-sequence fits trained on the first half of iterations and evaluated on the second half show high $R^2$ across vision (ResNet, ViT) and language (GPT-2), optimizers (SGD, AdamW, Muon-NSGD), and LR schedules.
- **Simple, actionable guidance.** Horizon-aware schedules with $\eta_{\text{peak}} \propto 1/\sqrt{T}$, paired with a few pilot runs to pick $\eta_{\text{ref}}^\star$, provide a practical recipe for early forecasting and planning.
- **Cross-horizon and cross-size extrapolation.** The paper demonstrates extrapolation up to $\sim 80 \times$ in horizon and $\sim 70\times$ in model size, and complements with analyses of external large-scale results.

**Weaknesses:**

- **Limited theoretical novelty.** Main results rely on standard convex SGD analysis; $\mathcal{O}(1/\sqrt{T})$ rates are classical. The main theoretical additions are schedule-wise constants (some via approximation).
- **Reproducibility and robustness gaps.** No code/logs are released. Experiments primarily vary LR and $T$; other key hyperparameters (weight decay, momentum/$\beta$’s, clipping, batch size) are mostly fixed (e.g., wd$=0.01$) with no ablations.
- **Scope of applicability.** Appendix~D shows the $1/\sqrt{T}$ law may fail for test loss under overfitting; diagnostics to detect onset/breakdown of the “convex-like” regime are not provided.

**Questions:**

- Will you release code, configurations, and logs (including seeds) for all experiments and figures?
- How sensitive are $(\tilde{q}\_1,\tilde{q}_2, L\_{\infty})$ and fit quality to weight decay, momentum/$\beta$’s, gradient clipping, batch size, data order, and seeds? Please add ablations with error bars.
- Can you justify Condition 2.4’s sum $\to$ integral replacement or provide bounds connecting the integral “exam” to the discrete last-iterate behavior?
- In your experiments, how sensitive are fitted slopes/intercepts to batch size?
- Can you propose diagnostics to detect the onset and breakdown of the “convex-like” regime (e.g., via curvature/Hessian measures) and validate test-loss predictions where possible?

**Details Of Ethics Concerns:**

The work uses standard datasets and literature-derived results; no ethical issues are apparent from the content provided.

---

> ### Author Response · Authors · 2025-11-18
>
> Thank you for reading our paper to details and the feedback! Here are point-to-point response. If you are satisfied with our response, we would be grateful if you could consider raising your score.
>
> *--- Main results rely on standard convex SGD analysis; O(1/sqrt(T)) rates are classical. The main theoretical additions are schedule-wise constants (some via approximation).*
>
> We agree that our theoretical additions are schedule-wise constants, on top of Defazio et al's Theorem 10 and Corollary 12. To our best knowledge, this result is relatively recent as dated in 2023, in the sense O(1/sqrt(T)) convergence is established on (1) any lr sequence and (2) last iterate. The more classical results are either on average iterate (as in equation 2.3), or on last iterate but specific lr sequence, such as $c/\sqrt{t}$ learning rate in Theorem 2 of https://arxiv.org/pdf/1212.1824.
>
> *---No code/logs are released. Will you release code, configurations, and logs (including seeds) for all experiments and figures?*
>
> Yes we will release code and logs in the camera-ready version. We have provided the configurations in Appendix B. Specifically, we use the default seed 1337 in line 106 of https://github.com/karpathy/nanoGPT/blob/master/train.py, which is a public codebase.
>
> Generally speaking, you can directly reproduce our results on nanoGPT codebase by launching runs with $1/\sqrt{T}$ scaled peak learning rate, without the need of our code, since our code only has minimal changes. Additionally, we have prepared a single ``train\_ICLRrebuttal.py'' in the supplementary material for plug-and-play into nanoGPT code. Here is an example: by setting multiple training horizons in $T=[60^2, 80^2, 100^2,120^2,140^2, 160^2, 180^2, 200^2]$, we run
>
> torchrun --standalone --nproc\_per\_node=8 train\_ICLRrebuttal.py --learning\_rate\_ref=0.3 --max\_iters=T
>
> This will give the loss (train and test) under cosine lr schedule with AdamW optimizer. We can observe $O(1/\sqrt{T})$ convergence of the last-iterate loss if we fit a linear regression of the loss against $1/\sqrt{T}$ for multiple T.
>
> *---Experiments primarily vary LR and
> ; other key hyperparameters (weight decay, momentum, clipping, batch size) are mostly fixed (e.g., wd) with no ablations. How sensitive are $q_1,q_2,L_\infty$ and fit quality to weight decay, momentum, gradient clipping, batch size, data order, and seeds? Please add ablations with error bars.*
>
> We didn't incorporate this ablation in our original submission, because our scope is the scaling law of loss and learning rate (while keeping other hyperparameters unchanged). According to our theory, $q_1,q_2,L_\infty$ will be sensitive to other hyperparameters, while **the fit quality is robust**. For example, our GPT experiments (Figure 8 to 10) uses no clipping and our VLM experiment (Figure 11) uses clipping threshold 1.0, but both have good linear fit; our SGD in Figure 2 uses no momentum and no weight decay and our AdamW in Figure 3 uses momentum and 0.001 weight decay, but both have good linear fit.
>
> We add **the requested ablation in Appendix D.3**. All fits have high R2 score (>0.99).
>
> *---Can you justify Condition 2.4’s sum integral replacement or provide bounds connecting the integral “exam” to the discrete last-iterate behavior?*
>
> Yes. The intuition is that integrals can be approximated by Riemann sum (also known as rectangle method) in numerical integration. Concretely, for constant learning rate, the discrete behavior is
> $$E L(w_T)\leq L_* + \frac{D^2}{2T\eta}+ \frac{G^2\eta}{2}+ \frac{G^2\eta}{2}\sum_{k=1}^{T-1}\frac{1}{T-k}$$
>
> And the continuous integrals give,
> $$L_* + \frac{D^2}{2T\eta}+ \frac{G^2\eta}{2}+ \frac{G^2\eta}{2}\ln T.$$
>
> Using the bound $\ln(T) - \frac{1}{T} \le\sum_{k=1}^{T-1}\frac{1}{T-k} \le\ln(T-1) + 1,$, we can show the gap is asymptotically 0. For other schedules like cosine decay, the derivation may be much more complicated and thus omitted.

---

> ### Author Response · Authors · 2025-11-18
>
> *---Can you propose diagnostics to detect the onset and breakdown of the “convex-like” regime (e.g., via curvature/Hessian measures) and validate test-loss predictions where possible?*
>
> We rely on empirical evidence to detect the onset of convex-like regime, that is, we launch multiple runs with increasing training horizon and fit linear regression in an online fashion. If the goodness of fit is high, then we have entered the regime.
>
> We didn't opt for curvature/Hessian measures because (1) it is difficult to estimate these measures on large model with more than 1M param, and (2) it is not clear what type of convexity is presented. This is discussed in Remark 2.2 that the standard convexity may not be necessary, and star-convexity is potentially sufficient. In this scenario, Hessian may not be informative. Some evidence that Hessian eigenvalues are mostly positive but some are still negative can be found in "An Investigation into Neural Net Optimization via Hessian Eigenvalue Density".
>
> We can empirically fit two linear regressions, one for train loss and the other for test loss, to inform on the overfitting regime. Unfortunately, it remains a mystery to us why language modeling is harder to overfit empirically than vision tasks.

---

### Author Response · Authors · 2025-11-18
**New results**

We sincerely thank all reviewers for their efforts and constructive advice. We have responded to each point and would like to hear back from the reviewers soon.

Below are some new experiments and results that further validate our generalizations.

1. As requested by Reviewer WNR5, we add **Table 7 in Appendix D.3 for the ablation study over other key hyperparameters**, such as weight decay, gradient clipping, momentum coefficient, batch size, and random seeds. The results are consistent with our Generalization 4 with R2 score $>0.99$  across different settings.

2. We add **Section 5.4 as multi-modal experiment with vision-language models**. This experiment is fine-tuning, in complement to our pre-training experiments. Our generalization indeed holds in this interesting task, not only because the task is difficult and new, but also because this model uses two different learning rates (one for backbone and the other for modality projector). We actually use $1/\sqrt{T}$ scaling on both learning rates (see Appendix B, last paragraph for details).

3. We add **Appendix D.2 as a experiment with language model fine-tuning**. This experiment includes parameter-efficient fine-tuning (LoRA), in complement to our full model training experiments. Our generalization still holds despite that over 99% of the parameters are frozen and hence not updated.

---

### Author Response · Authors · 2025-12-03
**Summary of rebuttal and new results**

Dear AC and reviewers,

We appreciate your efforts in handling our paper. We summarize our responses that address each weakness raised by reviewers (questions are answered in original responses as well). We also highlight some new experiments that further strengthen this paper.

1. Reviewer cgXX raises 28 weaknesses (15 important ones, some of which are about wording and typos; 4 unclear things; 9 presentation issues). We will only highlight the really important ones here.

* Weakness 1: "Line 185: “Section 2.4” is mentioned when, in fact, the paper has no “Section 2.4”. What is supposed to go there?"

**This is to our shock!** We checked our submission history and we clearly have Section 2.4 from line 201 to line 228 in our original submission (and the current one). This section is very important as it determines which learning rate schedule is qualified. Missing this may be a system error and may have unfairly resulted in this negative score. We sincerely hope the AC would consider this extreme adversity when making the decision.

* Weakness 2: "why derive equation (2.3) when at the end, as the paragraph that follows equation (2.4) states, only (2.4) is going to be used in the whole paper?"

We included (2.3) to be more self-contained and readable, so the authors can understand at least the first three terms in (2.4). However, we hope the reviewer would understand that it took a long analysis in that paper to derive the bound, and it is too lengthy to include in this work.

* Weakness 3: "I strongly suggest to put all generalization bounds on a single table to aid the reader to better understand the paper."

We have **added the requested table in Appendix A.2**.

2. Reviewer WNR5
 * Weakness 1: Limited theoretical novelty due to "Main results rely on standard convex SGD analysis; rates are classical."

We respectfully disagree as we rely on **non-classical results** in "Optimal Linear Decay Learning Rate Schedules and Further Refinements" (Theorem 10 and Corollary 12), which is dated in 2023 and relatively recent.

* Weakness 2: Reproducibility. "No code/logs are released..."

We kindly remind the reviewer that **all configurations are provided in Appendix B**, from which the results can be easily reproduced using the public codebase https://github.com/karpathy/nanoGPT/blob/master/train.py. Additionally, we have **prepared a single ``train_ICLRrebuttal.py'' in the supplementary material** for plug-and-play with a brief tutorial.

* Weakness 3: Robustness gaps. "other key hyperparameters (weight decay, momentum/’s, clipping, batch size) are mostly fixed ..with no ablations."

We argue that our experiments have shown that **the fit quality is robust** to different hyperparameters.

Furthermore, we add the **requested ablation over other key hyperparameters in Table 7 in Appendix D.3** , such as weight decay, gradient clipping, momentum coefficient, batch size, and random seeds. The results are consistent with our Generalization 4 with R2 score (>0.99) across different settings.

3. Reviewer yGzE only raises one weakness "is fitted coefficients fixed regardless of the maximal learning rate?"

We clarify that **we can predict the loss regardless of the maximal learning rate (without re-tuning)**, if the maximal learning rates are scaled by $1/\sqrt{T}$. **We confirm the practical benefit** that all we need are short runs on small models to tune $\eta_{ref}$ and then automatically determine the maximal learning rate for any long runs (up to 80 times in scale) and on large models (up to 70 times in scale).

4. Reviewer CJJk

* Weakness 1: "The paper assumes a uniform bound on the expected gradient, which in many case does not hold...which is probably the reason why in the experiment part the paper has to adopt the Adam optimizer instead of the regular SGD"

We have convinced the reviewer that **we may only need a bound on the optimization trajectory** ($\exists G$ s.t. $E|g(w_t)|^2 \leq G^2$ for $w_1,w_2,...w_T$), instead of a uniform bound ($\exists G$ s.t. $E|g(w)|^2 \leq G^2$ for all $w$). We have kindly corrected the reviewer that **we actually have SGD in Figure2 (ResNet model and ImageNet dataset)**.

* Weakness 2: "The paper can consider to incorporate several examples that explicitly apply this qualifying exam and validate the result using experiments."

**We kindly remind the reviewer that our Theorem 1 exactly addresses this** (discussed right after Condition 2.4 and proved in Appendix A). Concretely, we have applied this qualifying exam to 5 lr schedules and showed that "linear decaying, cosine decaying, and WSD schedules indeed pass the qualifying exam, whereas the constant and square-root inverse schedules fail."

5. **New results on LoRA finetuning and multi-modal vision-language model** further validate our scaling laws (see https://openreview.net/forum?id=dSdLqg02tx&noteId=5TXtMadmSH).

Hope everyone gets through this difficult submission cycle.

---

### Meta-Review · Area_Chair_FHWX · 2026-01-07

**Summary:**

This paper proposes a scaling law for the learning rate and loss of machine learning model training. The authors took motivation from the convex case and did empirical analysis to find a law to choose the step size for a non-convex, large-scale ML model. The experiment results are very promising. The reviewers have concerns and questions regarding the reproducibility, interpretations, and underlying theory for the results. Nevertheless, the practicality of the results is of utmost importance.

**Reviewer Concerns:**

Reviewer cgXX raised some questions; however, they did not seem to show any major weaknesses. However, the authors should fix the typos and implement the changes in writing as noted in the rebuttal.

Other comments are: Since the contribution of this paper is mostly empirical, the authors should include their reproducible code in the final version. Only describing experiments or stating the parameters does not seem to be sufficient. Also, more detailed ablation studies and studies on the scope of this scaling law (e.g., how model sizes generalize) would be beneficial.

**Reviewer Scores:**

There might be some chance that Reviewer cgXX will respond and change their score.

---

### Decision · Program_Chairs · 2026-01-26

Accept (Poster)